# Are High-Quality AI-Generated Images More Difficult for Models to Detect?

Yao Xiao [1]   Binbin Yang [1]   Weiyan Chen [1]   Jiahao Chen [2]   Zijie Cao [1]   Ziyi Dong [1]   Xiangyang Ji [3]   Liang Lin [1 4]
Wei Ke [2]   Pengxu Wei [1 4]

## Abstract

The remarkable evolution of generative models has enabled the generation of high-quality, visually attractive images, often perceptually indistinguishable from real photographs to human eyes. This has spurred significant attention on AI-generated image (AIGI) detection. Intuitively, higher image quality should increase detection difficulty. However, our systematic study on cutting-edge text-to-image generators reveals a counterintuitive finding: AIGIs with higher quality scores, as assessed by human preference models, tend to be more easily detected by existing models. To investigate this, we examine how the text prompts for generation and image characteristics influence both quality scores and detector accuracy. We observe that images from short prompts tend to achieve higher preference scores while being easier to detect. Furthermore, through clustering and regression analyses, we verify that image characteristics like saturation, contrast, and texture richness collectively impact both image quality and detector accuracy. Finally, we demonstrate that the performance of off-the-shelf detectors can be enhanced across diverse generators and datasets by selecting input patches based on the predicted scores of our regression models, thus substantiating the broader applicability of our findings. Code and data are available at GitHub.

## 1. Introduction

Recently, deep generative models have demonstrated impressive capabilities in generating photorealistic images from

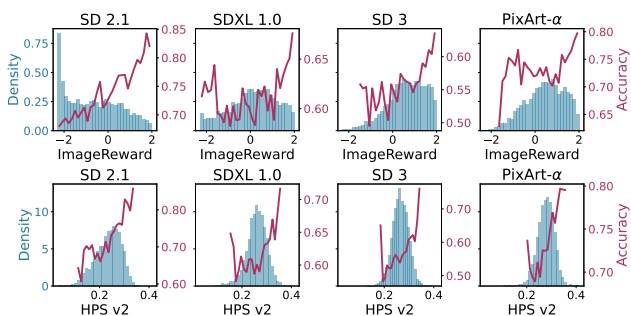

Figure 1: Average detector accuracy (the **red** curve) on generated images with different quality scores predicted by human preference models (ImageReward (Xu et al., 2023) and HPS v2 (Wu et al., 2023a)) and the distribution of quality scores (the **blue** histogram) for each generator. Counterintuitively, for the same generator, images with higher quality scores tend to be easier to detect.

input text prompts (Ramesh et al., 2021; Nichol et al., 2021; Rombach et al., 2022; Podell et al., 2024; Esser et al., 2024; Chen et al., 2024c). As the generated images become perceptually indistinguishable from real images to human eyes, they pose substantial threats to the spread of disinformation, particularly in the context of news dissemination related to political and social issues, and to the security risks of personal information.

Accordingly, AI-generated image (AIGI) detection aims to build a binary classification model for distinguishing real and fake (*i.e.*, AI-generated) images. While existing methods for AIGI detection (Ojha et al., 2023; Tan et al., 2024; Koutlis & Papadopoulos, 2024; Baraldi et al., 2024; Chen et al., 2024a) may achieve promising results on benchmark datasets (Zhu et al., 2023; Bammey, 2023; Baraldi et al., 2024; Chen et al., 2024a), recent studies (Yan et al., 2025; Cavia et al., 2024) reveal that their performance in real-world applications is unsatisfactory due to a mismatch in data distribution. Specifically, the AIGIs in existing datasets are randomly generated without ranking and filtering, while in the real world, synthetic images uploaded on the Internet are more likely of high quality and aligned with human preferences. This discrepancy in image quality between the training data and real-world cases seems to contribute to the

[1]School of Computer Science and Engineering, Sun Yat-sen University, Guangzhou, China [2]School of Software Engineering, Xi'an Jiaotong University, Xi'an, China [3]Department of Automation, Tsinghua University, Beijing, China [4]Peng Cheng Laboratory, Shenzhen, China. Correspondence to: Pengxu Wei <weipx3@mail.sysu.edu.cn>.

*Proceedings of the 42$^{nd}$ International Conference on Machine Learning*, Vancouver, Canada. PMLR 267, 2025. Copyright 2025 by the author(s).

suboptimal performance of detectors in practice, and raises an important question: *Are high-quality generated images preferred by humans more difficult for models to detect?*

To study the performance of existing AIGI detectors on high-quality generated images preferred by humans, we collect a dataset of high-quality images generated by text-to-image models with diverse prompts and obtain their quality scores predicted by human preference models (Xu et al., 2023; Wu et al., 2023a). By testing the average accuracy of six detectors, we observe that generated images with higher quality scores are consistently *easier* to detect across different generators and preference models, as depicted in Figure 1.

To explain this counterintuitive phenomenon, we analyze two primary factors of the quality score: the text prompts and image characteristics. Firstly, we observe that images generated from shorter prompts achieve higher quality scores, likely because existing generators struggle to follow complex prompts faithfully. More importantly, these images are also more easily spotted by AIGI detectors, which provides valuable insights into the observed phenomenon.

To further study the common characteristics of generated images with higher quality scores and higher detector accuracy, we conduct clustering-based analyses using quality-related features extracted by a human preference model (Xu et al., 2023). By examining representative image clusters, we find that certain low-level image characteristics, such as high saturation and rich texture, may serve as indicators of high quality scores and high accuracy for detectors. This motivates us to apply multiple linear regression to investigate the correlation between various image characteristics (independent variables) and both quality scores and detector accuracy (dependent variables). As expected, regression models for different dependent variables consistently highlight the influence of several low-level image characteristics. To demonstrate the applicability of our findings and examine the generalization of the regression models, we apply them to the input patch selection of off-the-shelf detectors and evaluate their performance across different generators and datasets. Experimental results reveal that selecting the most detectable patch identified by the regression models can enhance the performance of several existing detectors.

The main contributions of this paper are as follows:

- We collect a high-quality and diverse AIGI dataset to enable the study of the relationship between image quality and the detection difficulty.

- We reveal that high-quality AIGIs preferred by humans tend to be easier to detect for existing AIGI detectors.

- We investigate the correlation between average detector accuracy and quality scores predicted by human preference models, studying the influence of text conditions and image features. Our findings suggest that images generated from short text prompts or exhibiting specific low-level features (*e.g.*, high saturation and rich texture) tend to achieve higher quality scores while being easier to detect.

- We present a potential application of our findings in enhancing the detector performance: identifying the most detectable patch of an input image based on its low-level characteristics.

## 2. Related Works

### 2.1. AI-Generated Image Detection

**Methods.** In the common setup of AI-generated image (AIGI) detection, a detector is trained on fake images generated by one or more generators and paired real images. The detector is expected to generalize to fake images from unseen generators. Existing AIGI detection methods can be primarily categorized as *fingerprint-based* or *end-to-end*. Fingerprint-based models rely on certain low-level fingerprints of the generated images, such as diffusion reconstruction error (Wang et al., 2023), up-sampling artifacts (Tan et al., 2024), and filters proposed for steganalysis (Fridrich & Kodovsky, 2012; Zhong et al., 2024; Chen et al., 2024b). In contrast, end-to-end models classify an input image directly based on its pixel values (Wang et al., 2020). To improve the generalization of end-to-end models, some methods (Ojha et al., 2023; Koutlis & Papadopoulos, 2024) utilize features from large-scale pre-trained models such as CLIP (Radford et al., 2021). Others adopt contrastive learning to promote the separation of real and fake images in embedding space (Baraldi et al., 2024; Chen et al., 2024a).

**Benchmarks.** Several benchmarks are constructed to evaluate the generalization performance of AIGI detectors, featuring the coverage of diverse generators. ForenSynths (Wang et al., 2020) consists of images generated by 6 generative adversarial networks (GANs) (Goodfellow et al., 2014) and 5 other convolutional neural networks. GenImage (Zhu et al., 2023) is constructed based on the 1000 ImageNet classes (Deng et al., 2009), covering GANs and diffusion models. Specifically, for text-to-image diffusion models like Stable Diffusion (Rombach et al., 2022), a simple prompt template "photo of [class]" is applied to construct the class conditioning prompts. With the prevalence of text-to-image generators, recent works (Lu et al., 2023; Baraldi et al., 2024; Chen et al., 2024a) collect prompts for generation from the image captions in existing datasets, including CC3M (Sharma et al., 2018), LAION (Schuhmann et al., 2022), and COCO (Lin et al., 2014). Synthubuster (Bammey, 2023) obtains captions of real images from RAISE-1k (Dang-Nguyen et al., 2015) via Midjourney descriptor (Midjourney Inc., 2024) and CLIP Interroga-

tor (pharmapsychotic, 2022). Although the text prompts in these datasets may cover a wide range of semantics, they are relatively short and simple, in contrast to the long and descriptive captions utilized by more advanced text-to-image models (Betker et al., 2023; Chen et al., 2024c). This potentially limits the diversity of images generated from the same generator in terms of visual features and complexity.

**Detector performance on high-quality AIGIs.** Unlike images in previous benchmarks that are randomly generated, real-world AIGIs tend to be of higher quality due to more advanced generators (*e.g.*, Diffusion Transformers (Peebles & Xie, 2022; Esser et al., 2024; Chen et al., 2024c)), the application of prompt engineering (Liu & Chilton, 2022), and human selection of candidate generation outputs. To study existing detectors on real-world AIGIs, Cavia et al. (2024) collect real and fake images from social networks, and find that detectors trained on existing datasets generalize poorly on these real-world data. Similar conclusions are drawn by Yan et al. (2025), who construct a dataset of high-quality AIGIs collected from online communities and filtered by human annotators to ensure they are challenging for humans to detect. In addition, Song et al. (2024) argue that the generalization issue of deepfake detectors can be attributed to their reliance on easy-to-spot artifacts in low-quality training samples. However, the poor generalization of existing detectors on real-world AIGIs can also be explained by other factors, such as data biases concerning image size and compression (Grommelt et al., 2024; Ricker et al., 2024). Therefore, whether high-quality AIGIs are more difficult for detectors remains an open question.

## 2.2. Quality Assessment for AI-Generated Images

Despite the prevalence of Inception Score (IS) (Salimans et al., 2016) and Fréchet Inception Distance (FID) (Heusel et al., 2017) in image generation evaluation, these metrics may not be suitable for assessing the visual quality of individual images and are not well-aligned with human preference, as suggested by (Kirstain et al., 2023; Wu et al., 2023b). To this end, a series of human preference models (Kirstain et al., 2023; Xu et al., 2023; Wu et al., 2023b;a; Zhang et al., 2024b) are constructed based on large-scale AIGI datasets with human preference annotations, including ratings of single images or rankings of image pairs in the presence of corresponding text prompts.

## 3. Are Higher-Quality Images Harder to Detect as Intuitively Believed?

Intuitively, high-quality generated images preferred by humans should contain fewer visible artifacts and therefore may be more difficult to detect. In this section, we aim to answer the question: *Are high-quality generated images preferred by humans harder for models to detect, as intu-*

*itively believed?* To this end, we collect a high-quality and diverse AIGI dataset and empirically study how detector accuracy may relate to the quality scores predicted by human preference models.

### 3.1. Evaluation Setup

**Dataset collection.** To study the relationship between image quality and detector accuracy, a diverse dataset with a sufficient number of high-quality generated images is essential. However, as reviewed in Section 2.1, existing benchmark datasets for AIGI detection suffer from limitations in image quality and diversity. These shortcomings stem from the exclusion of more advanced generators such as Diffusion Transformers (Peebles & Xie, 2022; Esser et al., 2024; Chen et al., 2024c), a lack of prompt engineering considerations (Liu & Chilton, 2022), and reliance on low-complexity text prompts. To this end, we construct a high-quality and diverse dataset by 1) collecting real images from four source datasets; 2) obtaining 4,000 captions spanning a wide range of complexity from these real images; and 3) generating fake images using these captions as prompts based on text-to-image generators, *e.g.*, Stable Diffusion 2.1 (SD 2.1) (Rombach et al., 2022), Stable Diffusion XL 1.0 (SDXL 1.0) (Podell et al., 2024), Stable Diffusion 3 (SD 3) (Esser et al., 2024), and PixArt-$\alpha$ (Chen et al., 2024c). Specifically, negative prompts are applied during generation to improve image quality, and a set of positive modifiers are randomly sampled and appended to prompts for increased diversity in image characteristics. Details are presented in Appendix A.1.

**Human preference models.** Since annotating large-scale datasets with human preferences is challenging, we utilize pre-trained human preference models (*e.g.*, ImageReward (Xu et al., 2023) and Human Preference Score v2 (HPS v2) (Wu et al., 2023a)) as a substitute for scoring. This provides a scalable and consistent approach to assessing image quality in alignment with human judgments. These models take a generated image and its corresponding text prompt as inputs and predict a quality score for the image. Although the preference models commonly take image-text alignment into account for quality assessment, we empirically find that this does not affect our conclusions, as discussed in Appendix B.2.

**AIGI Detectors.** To ensure the reliability of our conclusions, we evaluate six existing open-source AIGI detectors that generalize well on our dataset. We use the average accuracy across these detectors as an indicator of how easily a set of generated images can be detected. Specifically, the selected detectors include NPR (Tan et al., 2024), RINE (Koutlis & Papadopoulos, 2024), CoDE (Baraldi et al., 2024), DRCT (Chen et al., 2024a) (including its Conv-B and CLIP variants), and SuSy (Bernabeu-Perez et al., 2024). These

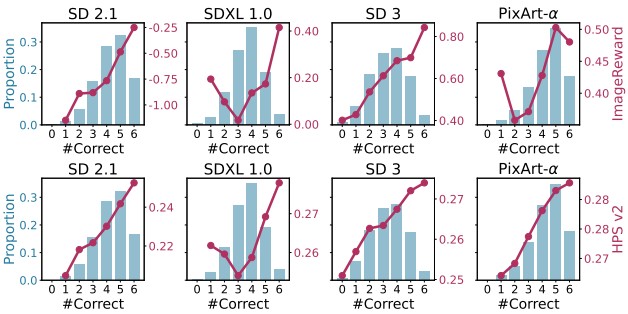

Figure 2: Average quality scores (the **red** curve) for generated images of different difficulty levels and the distribution of sample difficulty (the **blue** histogram). The difficulty is measured by the number of correct predictions (#Correct) from the 6 detectors. Notably, images that are easier to detect (*e.g.*, #Correct $\geq 5$) tend to have higher quality scores.

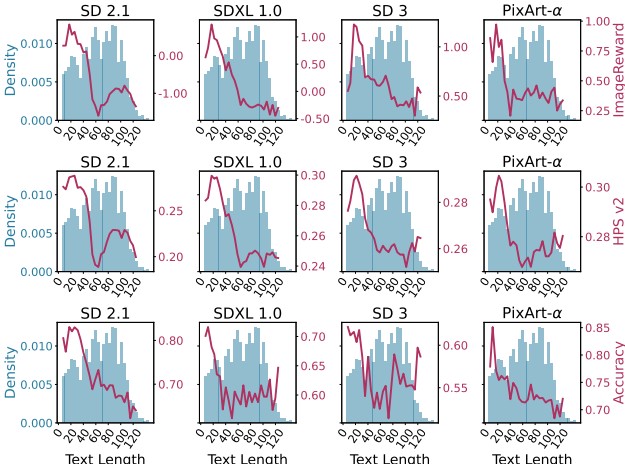

Figure 3: Quality scores (the **red** curves in the first two rows) and average detector accuracy (the **red** curves in the third rows) for images generated from prompts with different lengths, with the distribution of prompt lengths (the **blue** histogram). *Short* prompts ($\leq 20$ words) and *medium-length* prompts (21-40 words) generally correlate to higher quality scores and higher detector accuracy, in contrast with the *long* prompts ($> 40$ words).

detectors typically pre-process input images using center cropping without resizing. For evaluation, we directly apply the official pre-trained weights and configurations to testing on our dataset.

### 3.2. Main Observations

By investigating AIGI detectors on generated images of different quality scores, we make an intriguing observation: images with higher quality scores tend to be *easier* to detect, as suggested by Figure 1[1]. Specifically, for each generator and preference model, we uniformly split the range of quality scores into 30 segments and calculate the average detector accuracy on samples corresponding to each segment. A consistent trend emerges: **for images with quality scores above the distribution peak, average detector accuracy generally increases as the quality score rises**. Additionally, while detectors may achieve relatively high accuracy on images with low quality scores, this performance still falls short compared to that on the highest-quality images.

To further explore the relationship between quality scores and average detector accuracy, we plot the changes in quality scores as a function of detection difficulty in Figure 2, which can be viewed as a transposition of Figure 1. Specifically, since the possible values of average accuracy of the 6 detectors are discrete for a single sample, we use the number of detectors that correctly classify the sample as an index for detection difficulty (with a lower number indicating higher difficulty). As shown in Figure 2, **images that are easier to detect tend to have higher quality scores**. In conjunction with the observations in Figure 1, there is a positive correlation between quality scores and detector accuracy, particularly for images of higher quality.

---

[1]Additional results on more generators and human preference models are provided in Appendix B.1.

## 4. Why Do High Quality Scores Correlate With High Detector Accuracy?

To understand why higher quality scores of generated images may correlate with higher accuracy for detectors, we study the commonalities of high-quality images as assessed by human preference models. Specifically, as the quality scores predicted by these models depend on both the text prompt for generation and the features of the generated image, we first examine the effect of text prompt complexity in Section 4.1. Next, we explore the common characteristics of high-quality images through clustering analysis in Section 4.2. Finally, in Section 4.3, we conduct regression analyses to assess how these image characteristics correlate with quality scores and detector accuracy.

### 4.1. Influence of Text Prompt Complexity

As suggested by (Zhang et al., 2024a; Dong et al., 2024; Ma et al., 2024), existing text-to-image generators tend to struggle more with generating images from complex text prompts compared to short and simple ones. Hence, we have reason to believe that the quality of images generated under long and short text prompts differs, and we further validate this hypothesis through experiments. The results in Figure 3 confirm that images generated from *short* and *medium-length* prompts (at most 40 words) generally achieve higher average quality scores. Moreover, within this range, images generated from *short* prompts (at most 20 words) exhibit

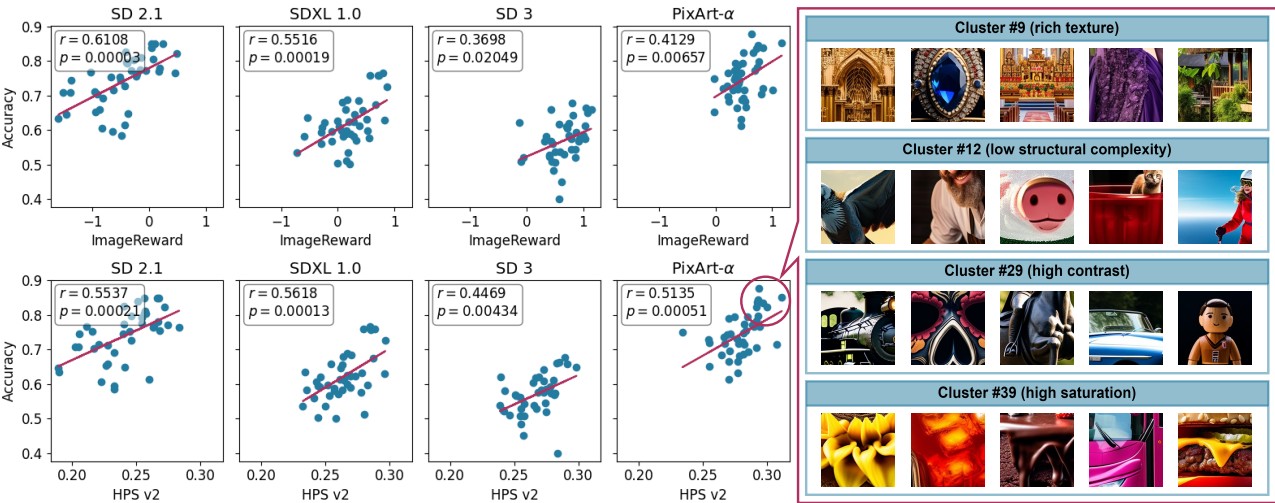

Figure 4: Analyses on generated images clustered by quality-related features extracted by ImageReward model. **Left**: The average detector accuracy and the average quality scores are consistently positively correlated at the cluster level. Each point represents an image cluster. The Pearson's correlation coefficient $r$ and the $p$-value are shown in the top left corner for each figure. **Right**: Visualization of image patches from typical clusters of PixArt-$\alpha$ with high average detector accuracy and high average quality scores. We take the center $224 \times 224$ patch in alignment with the pre-processing pipeline of detectors studied in this paper.

slightly lower quality scores compared to those generated from *medium-length* prompts with 21–40 words.

More importantly, we find that text prompt complexity has a similar influence on average detector accuracy. Figure 3 reveals that generated images produced from short prompts not only have higher quality scores but are also easier to detect. This finding provides further insight into our observation in Section 3, suggesting that certain types of generated images inherently exhibit both high quality scores and high detector accuracy.

To understand how text prompt complexity affects the generated images, we first conduct a preliminary analysis of their visual characteristics. A notable trait of images generated from short prompts is their relatively simple semantic content and structure, which can be attributed to the limited number of entities mentioned in the prompts. In addition, regarding the shortest prompts, we find that the absence of descriptions of visual attributes appears to contribute to the relatively lower quality scores of the generated images. In particular, prompts specifying object colors tend to produce images with higher saturation, which may lead to improved quality scores. Further details are provided in Appendix B.3.

While the analyses based on text prompt complexity offer valuable insights into the positive correlation between quality scores and average detector accuracy, grouping images solely by text length may not fully exploit the visual attributes that human preference models and detectors are sensitive to. Therefore, in the following sections, we extend our analysis through image clustering to better understand these underlying characteristics.

## 4.2. Image Clustering on Quality-Related Features

The observations in Section 4.1 suggest that high-quality images may exhibit certain characteristics that contribute to higher detector accuracy. To further explore these common traits, we cluster images based on their visual features and analyze the average quality scores and detector accuracy within each cluster. As we focus on quality-related features, the clustering is conducted in the feature space of the ImageReward (Xu et al., 2023) model. We apply the K-Means algorithm (Lloyd, 1982) with $k = 50$; other implementation details are presented in Appendix A.2.

Based on the clustering results, we first observe a significant positive correlation between average detector accuracy and quality scores at the cluster level, measured by the Pearson correlation coefficient, as shown in Figure 4, albeit the correlation is not strong. This trend further supports our observations in Figure 1 and Figure 2.

Then, we focus on clusters with both high average detector accuracy and quality scores. Through manual inspection of images within these clusters, we identify several recurring characteristics, including rich textures, low structural complexity, high contrast, and high saturation, as illustrated in Figure 4. We argue that these common characteristics may help explain the positive correlation between detector accuracy and quality scores. For example, highly saturated im-

Table 1: Comparison of regression models with different sets of independent variables (low-level or high-level features) and different dependent variable $y$. Standardized coefficients for each independent variable and the corresponding $R^2$-score are presented. The high-level features extracted by DINOv2 are reduced to 6 dimensions via PCA.

(a) Low-level image characteristics

| $y$ | Lightness | Contrast | Saturation | Sharpness | Texture Richness | Structural Complexity | $R^2$ |
|---|---|---|---|---|---|---|---|
| Accuracy | 0.10 | 0.52 | 0.43 | -0.33 | 0.42 | -0.30 | 0.77 |
| ImageReward | 0.26 | 0.20 | 0.34 | -0.46 | 0.35 | -0.25 | 0.81 |
| HPS v2 | 0.32 | 0.47 | 0.34 | -0.71 | 0.54 | -0.32 | 0.74 |

(b) High-level features (reduced to 6 dimensions by PCA)

| $y$ | $x_1$ | $x_2$ | $x_3$ | $x_4$ | $x_5$ | $x_6$ | $R^2$ |
|---|---|---|---|---|---|---|---|
| Accuracy | -0.18 | -0.23 | 0.20 | -0.23 | 0.04 | 0.11 | 0.76 |
| ImageReward | -0.39 | -0.22 | -0.04 | -0.18 | -0.02 | -0.12 | 0.76 |
| HPS v2 | -0.45 | -0.09 | 0.03 | -0.30 | -0.10 | -0.17 | 0.68 |

ages may be more visually appealing to humans, leading to higher quality scores, while their unnatural coloration could make them more distinguishable for detectors. Likewise, images with fine details and rich textures tend to receive higher quality scores, yet detectors might excel at identifying subtle artifacts in these intricate details, such as slight distortions along edges.

### 4.3. Linear Regression Analyses

To further substantiate the findings in Section 4.2 and quantitatively study how image characteristics may influence quality scores and average detector accuracy, we perform linear regression analyses on different features. Specifically, a linear regression model estimates the linear relationship between a dependent or target variable (*e.g.*, detector accuracy or quality score) and a set of independent or input variables (*e.g.*, image features). The linear relationship can be interpreted from the coefficients of independent variables learned from the data. The instantiations of the regression analyses and the results are detailed as follows.

**Independent variables.** Building on our previous observations, we first examine six low-level visual characteristics of images: lightness, contrast, saturation, sharpness, texture richness, and structural complexity. Each characteristic is quantified using a scalar metric, which we treat as an independent variable $x_i$ ($i = 1, 2, \cdots, 6$) in a linear regression model. The specific choices of metrics are detailed in Appendix A.3. Additionally, we explore whether high-level features also contribute to the observed positive correlation between average detector accuracy and quality scores. To this end, we extract high-level features from images using DINOv2 (Oquab et al., 2023), a self-supervised image encoder, and then reduce the feature dimension to 6 via PCA (Pearson, 1901) to align with the low-level features.

**Dependent variable.** The dependent variable $y$ can be (1) the average accuracy of the detectors, or (2) the average preference score predicted by the ImageReward or HPS v2 model. This yields three regression models for each set of features.

**Data fitting.** Building on the clustering analysis in Section 4.2, we perform regression analyses at the cluster level, averaging the metrics within each cluster. To obtain a more general regression model, we combine the cluster data for all four generators and fit the regression models to them. However, it should be noted that the average accuracy of detectors may vary considerably across different generators, as indicated by Figure 4. This variation can harm the regression models that fit the mixed data by considering only the aforementioned features as independent variables. To this end, we allow the regression model to learn generator-dependent intercepts to mitigate this bias. This is achieved by adding an indicator variable for each generator to represent the source of an image cluster. Since we are primarily interested in the relationship between independent variables $x_1, x_2, \cdots, x_6$ and the dependent variable $y$, rather than predicting the absolute value of $y$, the indicator variables and intercept can be ignored after the data fitting process. Additional details are provided in Appendix A.4.

**Results for low-level features.** Table 1a compares the standardized coefficients for each low-level image characteristic learned by the three regression models and presents the corresponding coefficient of determination ($R^2$). As suggested by the sign and magnitude of the coefficients, the image characteristics have a similar contribution to the prediction of accuracy and quality scores. Specifically, lightness, contrast, saturation, and texture richness of images positively correlate with both detector accuracy and quality scores, while sharpness and structural complexity have a negative correlation with these dependent variables. These conclusions are consistent with our observations and conjectures in Sections 4.1 and 4.2 and indicate the potential strengths and weaknesses of existing detectors. Moreover, the consistency across the three regression models reinforces the observed positive correlation between quality scores and average detector accuracy.

**Results for high-level features.** As suggested by Table 1b, not all principal components of the high-level features have consistent correlations with different dependent variables. Nevertheless, the subspace spanned by the unit vectors corresponding to $x_1$, $x_2$, and $x_4$ may provide certain support for the positive correlation between quality scores and average detector accuracy. As illustrated in Figure 5, clusters with lower values of $x_2$ and $x_4$ generally possess higher quality scores and are easier to detect. In addition, we notice that some outlier clusters (*e.g.*, #5, #16, #18) do not comply with the major trend. Specifically, Cluster #5 con-

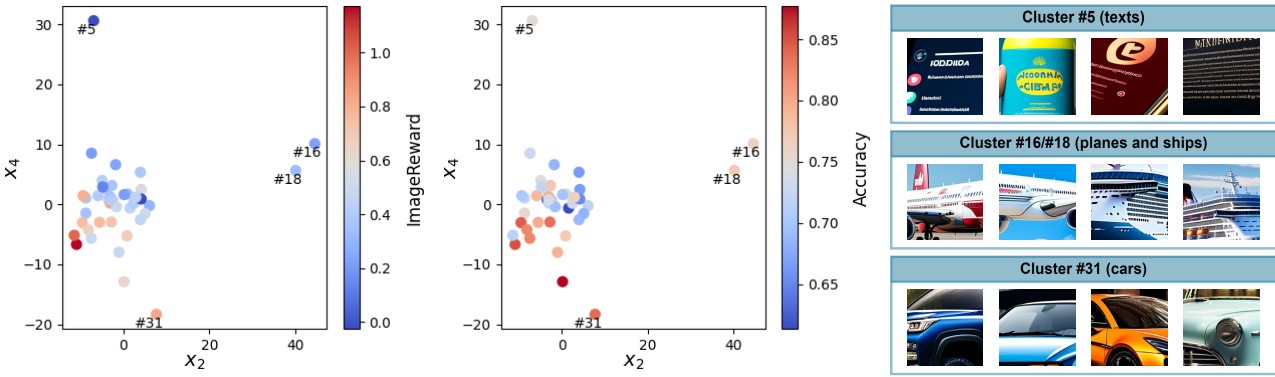

Figure 5: Analyses on clusters of images generated by PixArt-$\alpha$. **Left**: Cluster distribution in the high-level feature space. $x_2$ and $x_4$ correspond to the two independent variables in Table 1b and represent two principal components of the high-level features. Points are colored based on the ImageReward scores. **Center**: The same cluster distribution with points colored based on the average detector accuracy. **Right**: Visualization of image patches from outlier clusters.

sists of images with dense and distorted characters, which leads to low quality scores but makes them not difficult to detect. Similarly, detectors achieve relatively high accuracy on images in Clusters #16 and #18, which depict planes and ships with a number of portholes and have low quality scores. These examples suggest that existing detectors may be sensitive to certain semantic details like characters and portholes. This could explain the phenomenon observed in Figure 1 that some lower-quality images are also relatively easy to detect.

**Discussion.** The linear regression analyses suggest that certain low-level image characteristics (*e.g.*, high saturation or low sharpness) and high-level features tend to induce higher detector accuracy and higher quality scores. From a causal perspective (Pearl, 2009), these features could be the *confounders* underlying the counterintuitive positive correlation between accuracy and quality. However, through further intervention experiments presented in Appendix B.4, we find that several low-level image characteristics do not have uniform causal effects on the accuracy of different detectors, suggesting the existence of other confounding factors. Hence, for developing robust and generalizable AIGI detectors, future studies could benefit from further analyzing how these image features affect detectors.

## 5. Can We Optimize Patch-Based AIGI Detection Through Quality Analysis?

In this section, we aim to explore whether our analysis of image qualities in Section 4 can be leveraged to optimize existing patch-based AIGI detection methods, which typically rely on center-cropping and disregard other patches. Specifically, we examine whether the regression models from Section 4.3 can identify the most detectable patch in an image. To this end, we improve patch-based detectors by

selecting input patches according to the regression models' predictions and investigate whether their performance can be enhanced on various generators and datasets.

### 5.1. Selection Strategies for Input Patches

Most existing methods for AI-generated image detection assume that fake images are completely generated by models rather than locally edited from real images. Under this setting, a series of methods (Ojha et al., 2023; Tan et al., 2024; Koutlis & Papadopoulos, 2024; Baraldi et al., 2024; Chen et al., 2024a) adopt center cropping in their image pre-processing pipeline without resizing. While limiting the receptive field to a patch of size $224 \times 224$ may lead to substantial losses of high-level information, especially for high-resolution images, it preserves fine-grained details and enables detectors to spot low-level traces of generation (Corvi et al., 2023; Gragnaniello et al., 2021).

Based on our findings in Section 4, we hypothesize that selecting input patches according to the common characteristics of high-quality or easy-to-detect images can enhance the performance of existing detectors. Specifically, we first patchify the input image and then use one of the regression models from Section 4.3 to rank the patches by their predicted scores. By selecting the patch with the highest score for a given image, we expect to make the detection more accurate, particularly for detecting fake images.

However, this patch selection strategy may not have the same expected effects on different types of detectors, as reviewed in Section 2.1. Specifically, a recent study (Chen et al., 2024b) suggests that simple patches with poor texture are more effective for fingerprint extraction than complex patches. This may contradict the role of texture richness in our patch selection strategy based on the regression results. Consequently, we speculate that for fingerprint-based mod-

Table 2: Average detector accuracy (%) of end-to-end models on fake images. We compare different patch selection strategies based on regression models with different dependent variables $y$.

| Dependent Variable $y$ | Strategy | GenImage | DRCT-2M | | | | | Synthbuster | | | | Average |
|---|---|---|---|---|---|---|---|---|---|---|---|---|
| | | Midjourney | SDXL-Refiner | SDXL-Turbo | LCM-SDXL | SDv2-Ctrl | SDXL-Ctrl | DALL·E 2 | DALL·E 3 | Firefly | Midjourney | |
| / | Center | 76.3 | 75.3 | 56.8 | 62.6 | 75.0 | 76.2 | 38.1 | 51.4 | 41.4 | 75.0 | 62.8 |
| Accuracy | Easiest | **75.0** | **76.0** | **56.6** | **62.1** | **74.8** | **76.0** | **38.6** | 51.3 | **45.4** | 75.5 | **63.1** |
| | Random | 70.6 | 72.0 | 56.1 | 61.1 | 73.6 | 75.0 | 36.7 | 50.9 | 37.8 | 71.1 | 60.5 |
| | Hardest | 63.5 | 66.4 | 54.9 | 60.9 | 71.8 | 73.4 | 34.2 | 47.8 | 29.5 | 63.9 | 56.6 |
| ImageReward | Easiest | 73.1 | 75.0 | 56.2 | 60.5 | 74.4 | 76.0 | 37.6 | 50.1 | **44.6** | 70.9 | **61.8** |
| | Random | 70.6 | 72.0 | 56.1 | 61.1 | 73.6 | 75.0 | 36.7 | **50.9** | 37.8 | 71.1 | 60.5 |
| | Hardest | 68.6 | 69.3 | 54.9 | **62.4** | 72.2 | 73.6 | 35.7 | 50.8 | 37.2 | **71.5** | 59.6 |
| HPS v2 | Easiest | **74.1** | **75.7** | **56.3** | 61.0 | **74.5** | **76.0** | **38.6** | 51.0 | 44.2 | 73.7 | **62.5** |
| | Random | 70.6 | 72.0 | 56.1 | 61.1 | 73.6 | 75.0 | 36.7 | 50.9 | 37.8 | 71.1 | 60.5 |
| | Hardest | 66.3 | 67.3 | 55.0 | **62.0** | 72.1 | 73.5 | 34.3 | 49.7 | 34.5 | 69.2 | 58.4 |

els, selecting the patch with the lowest score predicted by the regression models may yield better detection performance.

In the following part of this section, we verify our hypotheses by experimentally comparing four input patch selection strategies: (1) **Easiest**: selecting the patch with the *highest* predicted score; (2) **Hardest**: selecting the patch with the *lowest* predicted score; (3) **Random**: selecting a random patch; (4) **Center**: taking the center patch, as in the aforementioned existing methods.

### 5.2. Experiments

**AIGI detectors.** In addition to the six AIGI detectors described in Section 3.1, we consider SSP (Chen et al., 2024b), which takes a single simple patch for fingerprint extraction. We train the SSP model on GenImage (Zhu et al., 2023) instead of using the official checkpoints with unsatisfactory generalization. Details are provided in Appendix A.5. Overall, the detectors can be categorized into: (1) **fingerprint-based models**: NPR (Tan et al., 2024) and SSP; (2) **end-to-end models**: RINE (Koutlis & Papadopoulos, 2024), CoDE (Baraldi et al., 2024), DRCT (Chen et al., 2024a), and SuSy (Bernabeu-Perez et al., 2024).

**Datasets.** To examine the generalization of the regression models on unseen generators, we test the detectors using different patch selection strategies on images generated by the following generators from existing benchmark datasets: Midjourney from GenImage (Zhu et al., 2023); SDXL-Refiner, SDXL-Turbo, LCM-SDXL, SDv2-Ctrl, and SDXL-Ctrl from DRCT-2M (Chen et al., 2024a); DALL·E 2, DALL·E 3, Firefly, and Midjourney from Synthbuster (Bammey, 2023). We clarify the data selection in Appendix A.6.

**Comparison of different regression models.** Table 2 shows the average accuracy of end-to-end detectors on fake images produced by different generators. **First**, selecting the input patch with the highest scores predicted by regression models (i.e., the Easiest strategy) generally provides higher average detector accuracy on various generated images, compared to the Random or Hardest strategy. **Second**, the Center strategy adopted by existing methods is comparable to the Easiest strategy and superior to the Random strategy, which possibly suggests the particularity of the center areas of generated images. **Third**, the three regression models provide consistent effects on average detector accuracy, although the two models fitting quality scores may be less effective in predicting the easiest patch. These results further validate the relationship between quality scores and average detector accuracy.

**Comparison of different detectors.** To study how different detectors are affected by patch selection strategies, we compare their average performance across all generators in Table 3. The Easiest and Hardest strategies are based on the regression model fitting accuracy. **First**, all end-to-end detectors except for DRCT (CLIP) achieve the best AP when selecting the input patch based on the Easiest strategy. **Second**, for DRCT (CLIP), we notice that the accuracy on real images is significantly lower than that of other end-to-end detectors. Adopting the Easiest strategy leads to a further decline in this metric, which indicates that this detector may learn different features for discriminating real and fake images. **Third**, the patch selection strategies have opposite effects on fingerprint-based detectors, where the Hardest strategy consistently provides better performance than the Easiest and Random strategies. For SSP, its original strategy of selecting the simplest patch can produce higher accuracy on fake images, but at the cost of lower accuracy on real ones. **Overall**, the results validate that the proposed patch selection strategies can be used to enhance the performance of some off-the-shelf detectors, while the "easiest" patches for end-to-end detectors may be the "hardest" patches for fingerprint-based detectors and vice versa.

Table 3: Average performance (%) across all generators for different detectors using different patch selection strategies. ACC: accuracy on all images; AP: average precision; $\text{ACC}_{fake}$/$\text{ACC}_{real}$: accuracy on fake/real images. (*: the strategy proposed in (Chen et al., 2024b) for SSP, which selects the simplest patch with the lowest texture diversity in the input image.)

| Type | Detector | Strategy | ACC | AP | $\text{ACC}_{fake}$ | $\text{ACC}_{real}$ |
|---|---|---|---|---|---|---|
| End-to-end | RINE | Center | **75.0** | 90.2 | **51.6** | 98.5 |
| | | Easiest | 74.2 | **91.9** | 49.7 | 98.9 |
| | | Random | 71.4 | 90.3 | 44.0 | 98.9 |
| | | Hardest | 67.6 | 90.0 | 36.4 | **98.9** |
| | CoDE | Center | 79.4 | 78.8 | 60.1 | 98.8 |
| | | Easiest | **79.7** | **79.2** | **60.8** | 98.7 |
| | | Random | 79.4 | 79.0 | 59.9 | **99.0** |
| | | Hardest | 77.8 | 77.2 | 56.9 | 98.7 |
| | SuSy | Center | 76.0 | 87.6 | 59.7 | 92.3 |
| | | Easiest | **77.8** | **89.4** | **62.1** | **93.6** |
| | | Random | 74.4 | 87.0 | 55.9 | 93.0 |
| | | Hardest | 69.8 | 82.7 | 48.6 | 91.1 |
| | DRCT (Conv-B) | Center | 75.4 | 84.8 | 53.7 | 97.1 |
| | | Easiest | **76.4** | **87.6** | 53.8 | **99.0** |
| | | Random | 75.1 | 84.2 | **54.1** | 96.0 |
| | | Hardest | 73.3 | 82.5 | 53.6 | 93.1 |
| | DRCT (CLIP) | Center | 78.9 | 88.4 | 89.0 | 68.9 |
| | | Easiest | 77.6 | 86.7 | **89.2** | 65.9 |
| | | Random | 78.4 | 87.6 | 88.5 | 68.3 |
| | | Hardest | **80.3** | **88.6** | 87.7 | **72.9** |
| Fingerprint-based | NPR | Center | 71.6 | 75.9 | 66.8 | 76.3 |
| | | Easiest | 69.4 | 74.6 | 66.6 | 72.1 |
| | | Random | 73.7 | 76.3 | 68.5 | 78.9 |
| | | Hardest | **76.6** | **78.6** | **71.1** | **82.2** |
| | SSP | Center | 72.0 | 79.7 | 51.7 | 92.4 |
| | | Easiest | 70.9 | 77.4 | 53.8 | 88.0 |
| | | Random | 73.4 | 81.0 | 54.9 | 92.0 |
| | | Hardest | 74.5 | **81.0** | 56.4 | **92.5** |
| | | Simplest* | **74.5** | 78.3 | **64.2** | 84.7 |

## 6. Conclusion

In this paper, we make a counterintuitive observation that high-quality AI-generated images (AIGIs) preferred by humans tend to be less difficult for models to detect. We explain the positive correlation between detector accuracy and the quality scores of generated images by investigating the influence of text prompts and image features. Our results suggest that images generated from shorter text prompts or those exhibiting specific low-level image characteristics (such as high saturation, high contrast, rich texture, and low structural complexity) correlate with higher quality scores and higher detector accuracy. Finally, we suggest that existing patch-based AIGI detectors can be improved by selecting the most detectable patch identified based on low-level visual characteristics. This validates the broader applicability of our findings.

## Acknowledgements

This work is supported in part by National Natural Science Foundation of China (NSFC) under Grant No.62376292, U21A20470, Guangdong Provincial General Fund No. 2024A1515010208.

## Impact Statement

This paper studies AI-generated image detection, which has become an important topic for AI governance as AI-generated images become more perceptually indistinguishable from real ones. The findings in this paper suggest potential strengths and weaknesses of existing detectors, and are expected to promote the development of more advanced models for detecting various AI-generated images. A potential negative impact lies in the facilitation of adversarial evasion attacks against existing detectors. For instance, as suggested by our results, lowering the lightness and contrast of generated images may impair the performance of certain kinds of detectors. A preliminary exploration of the effectiveness of such adversarial image manipulations is presented in Appendix B.4. However, we believe that this potential risk also motivates future research on robust detectors for AI-generated images in the real world.

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

# A. Implementation Details

## A.1. Dataset Collection

Following the pipeline of the construction of existing datasets for AIGI detection (Zhu et al., 2023; Bammey, 2023; Baraldi et al., 2024; Chen et al., 2024a), we collect real images from existing datasets, acquire the corresponding captions, and generate the fake images via a set of text-to-image models taking these captions as text prompts.

**Real image collection.** The contents of the fake images are loosely determined by the real images. Hence, to ensure the diversity of the dataset, we collect real images from four existing datasets: COCO (Lin et al., 2014), CC3M (Sharma et al., 2018), LAION-Aesthetic (Schuhmann et al., 2022), and SA-1B (Kirillov et al., 2023). In particular, SA-1B is included as it contains more images with complex scenes compared to the other datasets.

**Text prompt collection.** The text prompts are expected to be faithful descriptions of the visual content of the real images and cover various levels of complexity. To this end, we apply BLIP-2 (Li et al., 2023) and a large multimodal model (InternVL2-8B (Chen et al., 2024d)) to the captioning of real images. Specifically, we use different instructions for InternVL2-8B to obtain captions with varying complexity. We also include the high-quality captions from COCO and those produced for SA-1B by (Chen et al., 2024c), which supplement the low-complexity and high-complexity captions, respectively. After filtering and balancing the distribution of text lengths, we obtain 4,000 captions and use them as the text prompts for fake image generation. As depicted by the histogram in Figure 3, these prompts effectively cover a wide range of text length, from less than 10 words to over 120 words.

**Fake image generation.** The fake images are generated by six advanced open-source text-to-image models, namely Stable Diffusion 2.1 (SD 2.1) (Rombach et al., 2022), Stable Diffusion XL 1.0 (SDXL 1.0) (Podell et al., 2024), Stable Diffusion 3 (SD 3) (Esser et al., 2024), PixArt-$\alpha$ (Chen et al., 2024c), FLUX.1 [dev] (Black Forest Labs, 2024), and Infinity (Han et al., 2024). For each text prompt, we produce a fake image by each generator. To improve the quality of generated images and increase the diversity of image characteristics, we follow the prompt engineering adopted by (Baraldi et al., 2024). Specifically, a fixed negative prompt is applied for all images, while the positive modifiers are randomly sampled from a set of common modifiers in the generation of each image. The generated images are compressed by the same format and quality as the real counterparts to eliminate the compression bias (Grommelt et al., 2024).

## A.2. Image Clustering Based on Quality-Related Features

To obtain the quality-related features for image clustering, we utilize the ImageReward (Xu et al., 2023) model. An empty text is fed to the model instead of the text prompt corresponding to the input image, as we find that using the corresponding prompts can lead to a near-linear distribution of samples in the feature space, possibly due to the strong features related to image-text alignment. Real images corresponding to the 4,000 text prompts are included in the clustering process to indicate the potential discrepancy between real and fake images and are neglected in the subsequent analyses. However, we find that most clusters are mixed with real and fake images, which suggests the high fidelity of the generated images in our dataset. In addition, considering that the detectors studied in this paper mostly adopt center-cropping without resizing in their image pre-processing pipeline, we perform feature extraction based on the center $224 \times 224$ patches of images. Finally, to ensure the reliability of our analyses, we filter out the clusters with the number of generated images fewer than 30.

## A.3. Metrics for Low-Level Image Characteristics

- **Lightness**: Convert the image to HSV space, and take the average value of the Value channel.

- **Contrast**: Compute the root-mean-square (RMS) contrast (Peli, 1990), *i.e.*, the standard deviation of the pixel intensities.

- **Saturation**: Convert the image to HSV space, and take the average value of the Saturation channel.

- **Sharpness**: Apply the Laplacian operator on the grayscale image, and then compute the variance.

- **Texture richness**: Apply the Canny edge detector (Canny, 1986) to the grayscale image, and then compute the edge density, *i.e.*, the ratio of edge pixels.

- **Structural complexity**: Motivated by (Shen et al., 2024), we take the square root of the number of image segments detected by the SAM 2 model (Ravi et al., 2024).

### A.4. Multiple Linear Regression

In Section 4.3, we analyze the correlation between image features (independent variables) and the average detector accuracy or quality scores (dependent variable). Both independent variables $x_1, x_2, \cdots, x_6$ and the dependent variable $y$ are standardized to obtain the standardized coefficients (or beta coefficients) for comparison. The regression models are fit to the data using the least squares approach.

### A.5. Training SSP on GenImage

In our experiments, we find that the official checkpoints of the SSP model (Chen et al., 2024b) trained on fake images produced by single generators of GenImage (Zhu et al., 2023) fail to yield satisfactory results on other datasets. To obtain a more generalizable SSP model, we train it on a mixture of fake images generated by all generators, and eliminate the compression bias (Grommelt et al., 2024) in GenImage by aligning the compression format and quality of the fake images with the real ones.

### A.6. Test Data Selection

In Section 5.2, we test the detectors on fake images generated by various generators from multiple benchmark datasets (Zhu et al., 2023; Bammey, 2023; Chen et al., 2024a). To ensure that the results are meaningful for the validation of our findings, we filter out generators that satisfy any of the following conditions: (1) the generator is SD 2.1 or SDXL, which are included in our dataset and should not be used for testing the generalization of our regression models; (2) the generator is included in or very similar to the common generators used for the training of the detectors (*e.g.*, SD 1.4/1.5), since detectors consistently have high accuracy on images generated by seen generators; (3) the size of the generated images are $256 \times 256$ or smaller, leaving no room for patch selection.

## B. More Experimental Results and Further Discussions

### B.1. Validation on More Generators and Human Preference Models

Figure 1 and Figure 2 in the main text demonstrate our main observations concerning the relation between the average detector accuracy and image quality scores based on 4 generators (*i.e.*, SD 2.1, SDXL 1.0, SD 3, and PixArt-$\alpha$) and 2 human preference models (*i.e.*, ImageReward and HPS v2). To further validate that the observations are consistent across different generators and preference models, we extend Figures 1 and 2 with two more generators (a commercial DiT model, **FLUX.1 [dev]** (Black Forest Labs, 2024) and an autoregressive model, **Infinity** (Han et al., 2024)), and one additional human preference model (**MPS** (Zhang et al., 2024b)), as shown in Figures 6 and 7. Furthermore, we examine whether similar trends can be observed on user-created images from advanced closed-source generators, including **Midjourney v6** (Midjourney Inc., 2024) and **DALL·E 3** (Betker et al., 2023). To this end, we sample 50,000 and 75,000 generated images from existing datasets collected by Cortex Foundation (2024) and Egan et al. (2024), respectively. The evaluation results are presented in Figures 8 and 9. These results collectively validate that our main observations in Section 3 are consistent across a broad range of generators and different human preference models.

### B.2. Reducing the Influence of Image-Text Alignment in Image Quality Assessment

Existing human preference models (Xu et al., 2023; Wu et al., 2023a; Zhang et al., 2024b) commonly input the text prompt for generation in addition to the generated image, because a high-quality image in practice should be not only visually appealing but also aligned with the text prompt (*i.e.*, satisfying the intention of the user). However, this paper focuses more on the visual quality of the generated images, regardless of how the images are aligned with the text prompts.

To validate that the observed correlation between the detector accuracy and human preference scores can be attributed to the visual quality of images, we try to reduce the influence of the image-text alignment in the comparison of image quality by replacing the text input for the preference models. Specifically, instead of feeding the preference models with the original prompt for generation, we use the **BLIP-2 (Li et al., 2023) caption of the generated image itself**, which is expected to be well-aligned with the image as evaluated by the preference models. The results presented in Figures 10 and 11 are consistent with those produced by the standard metrics (*i.e.*, Figures 6 and 7), validating the correlation between the detector accuracy and image visual quality.

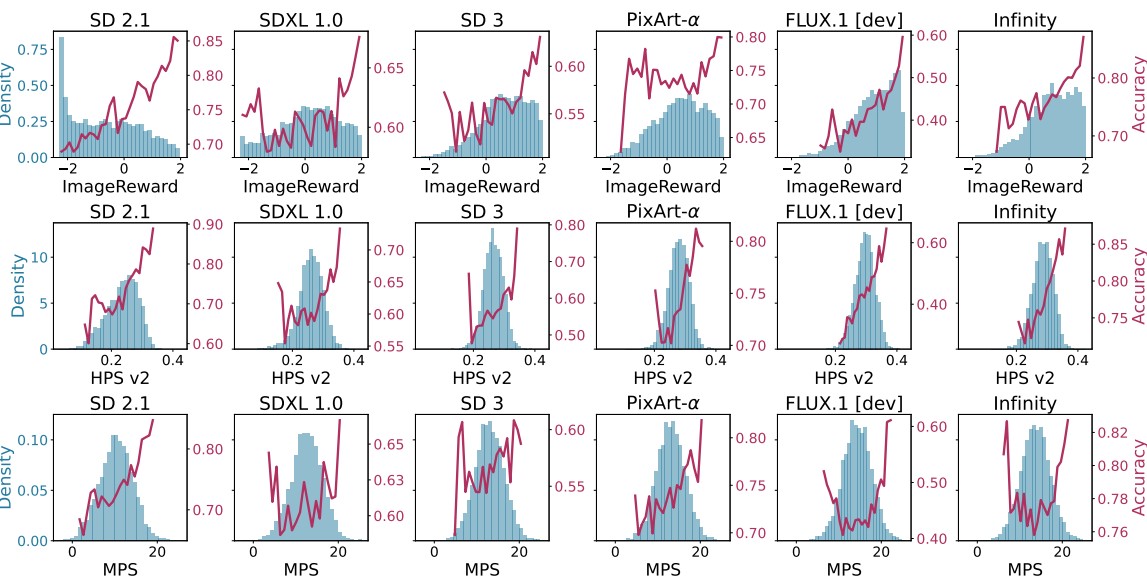

Figure 6: Average detector accuracy (the red curve) on generated images with different quality scores predicted by human preference models (ImageReward, HPS v2, and MPS) and the distribution of quality scores (the blue histogram) for each generator.

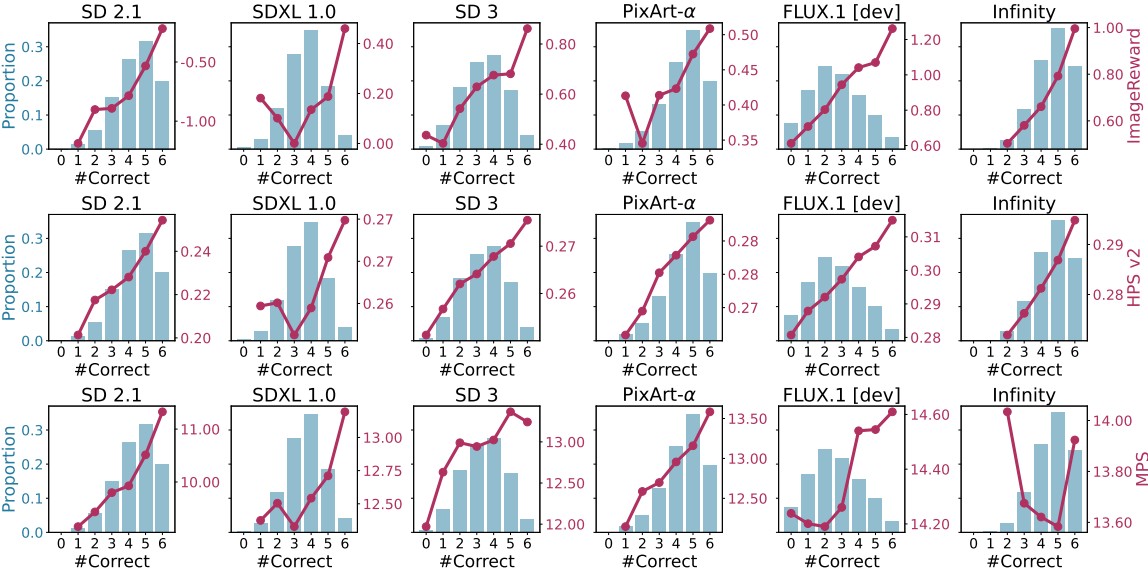

Figure 7: Average quality scores (the red curve) for generated images of different difficulty levels and the distribution of sample difficulty (the blue histogram). The difficulty is measured by the number of correct predictions (#Correct) from the 6 detectors.

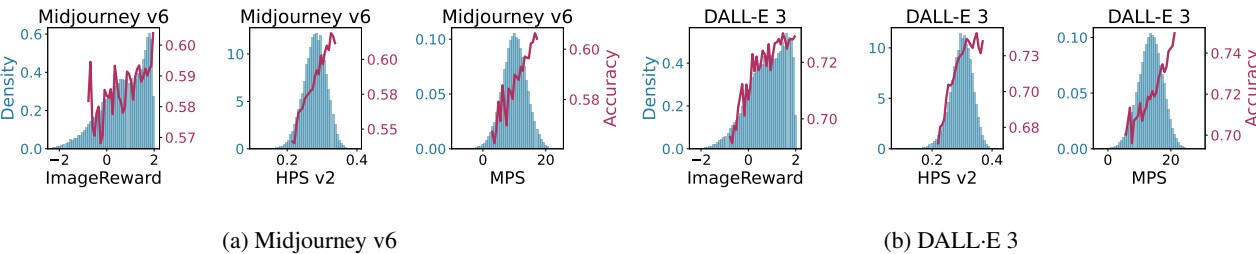

(a) Midjourney v6                                           (b) DALL·E 3

Figure 8: Average detector accuracy (the **red** curve) on generated images with different quality scores predicted by human preference models (ImageReward, HPS v2, and MPS) and the distribution of quality scores (the **blue** histogram) for **closed-source generators**.

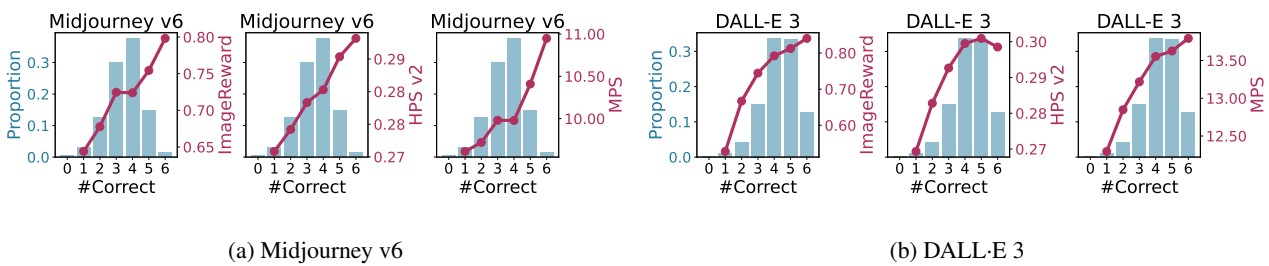

(a) Midjourney v6                                           (b) DALL·E 3

Figure 9: Average quality scores (the **red** curve) for generated images of different difficulty levels and the distribution of sample difficulty (the **blue** histogram) for **closed-source generators**. The difficulty is measured by the number of correct predictions (#Correct) from the 6 detectors.

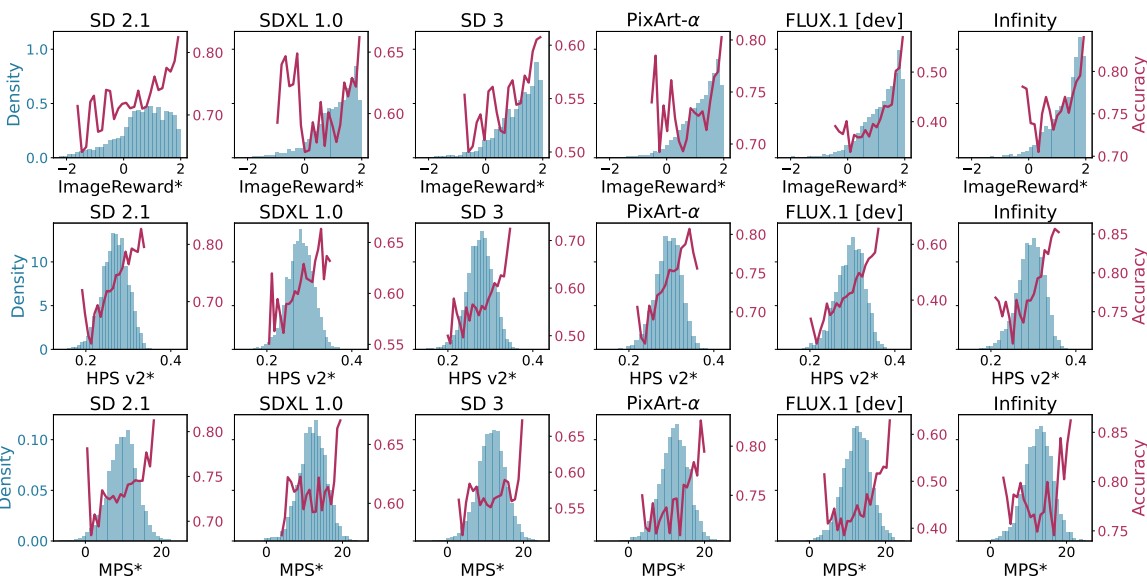

Figure 10: Average detector accuracy (the **red** curve) on generated images with different quality scores predicted by human preference models (ImageReward, HPS v2, and MPS; **\*: using the BLIP-2 caption of the generated image instead of the original prompt as the text input**) and the distribution of quality scores (the **blue** histogram) for each generator.

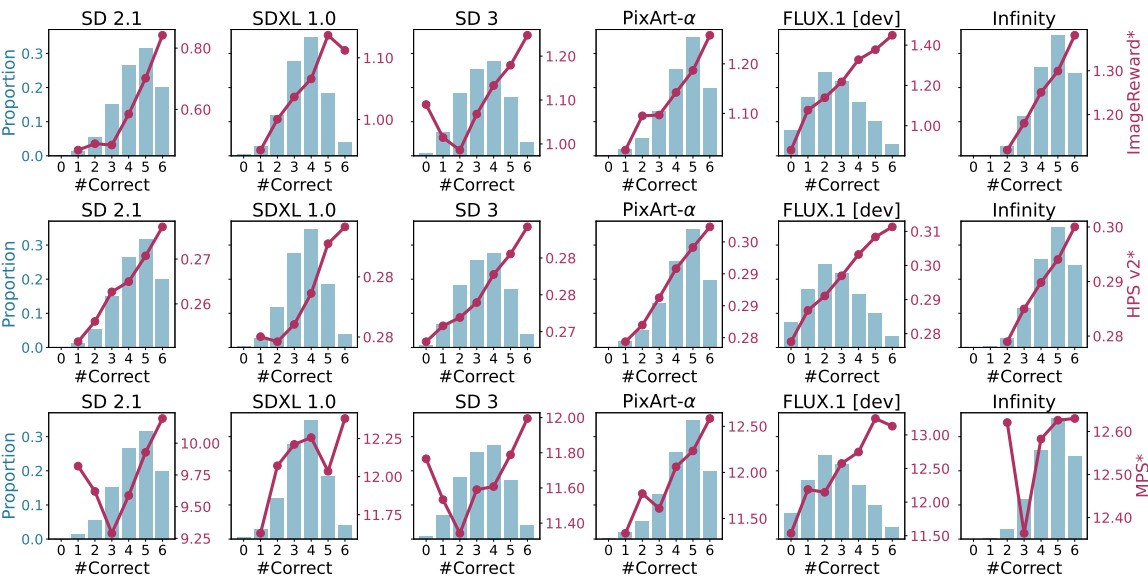

Figure 11: Average quality scores (the red curve) for generated images of different difficulty levels (**\*: using the BLIP-2 caption of the generated image instead of the original prompt as the text input**) and the distribution of sample difficulty (the blue histogram). The difficulty is measured by the number of correct predictions (#Correct) from the 6 detectors.

### B.3. Explaining the Influence of Text Prompt Complexity

In Section 4.1, we observe that (1) images generated from *short* and *medium-length* prompts (with at most 40 words) tend to have higher quality scores and higher average detector accuracy, as compared with those generated from longer prompts; (2) the quality scores are relatively lower for images generated from the *short* prompts (with at most 20 words) compared to those from *medium-length* prompts (with 21-40 words).

To explain observation (1), we speculate that images generated from shorter prompts tend to have lower structural complexity (which correlates with higher detector accuracy and higher quality scores as suggested in Section 4) as fewer entities are mentioned in the prompts. This can be validated by the results in Figure 12a.

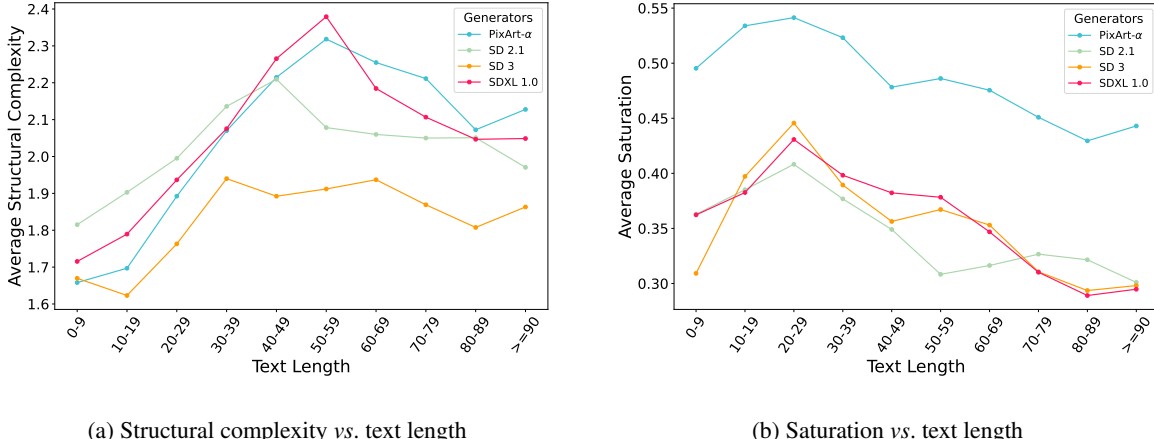

(a) Structural complexity *vs*. text length    (b) Saturation *vs*. text length

Figure 12: Average structural complexity and average saturation of images corresponding to different ranges of text length.

To explain observation (2), we first notice that the shortest prompts (*e.g.*, those with less than 10 words) may contain fewer descriptions of the attributes of the objects, as evidenced in the examples in Appendix C. Besides, images generated from these prompts generally exhibit lower saturation, as shown in Figure 12b. Therefore, we hypothesize that the lack of descriptions of visual attributes, especially those related to the colors, may lead to lower saturation of the generated image, and thereby lower quality scores and detector accuracy, as suggested in Section 4. We verify this hypothesis by counting the ratios of prompts with color-related descriptions within different ranges of text length, and comparing the average saturation of images generated from prompts with or without such descriptions, as shown in Table 4. It is clear that a higher ratio of prompts with text length in 20-29 have color-related descriptions, and images generated from prompts with such descriptions exhibit higher saturation on average.

Table 4: Comparison of text prompts with and without color-related descriptions. The prompt ratio is counted for three segments of text lengths, and the average saturation is computed over all samples with text length fewer than 30.

| Color-Related Descriptions | Prompt Ratio (%) | | | Average Saturation |
|---|---|---|---|---|
| | 0-9 | 10-19 | 20-29 | |
| With | 14.2 | 37.9 | 63.2 | 0.47 |
| Without | 85.8 | 62.1 | 36.8 | 0.40 |

### B.4. Adversarial Attacks Inspired by the Linear Regression Analyses

The linear regression analyses in Section 4.3 indicate the potential strengths and weaknesses of existing AIGI detectors, which could facilitate the understanding and promote the development of AIGI detection methods. However, the conclusions could also inspire adversarial attacks against existing AIGI detectors. Specifically, Table 1a suggests certain adversarial directions for image manipulations, such as decreasing the lightness, contrast, and saturation, or increasing the sharpness. To investigate the effectiveness of such adversarial directions, we implement these manipulations with different factors (0.5 means decreasing by 50%; 1.5 means increasing by 50%). The results in Table 5 suggest that: (1) the adversarial lightness and contrast manipulations are effective for DRCT and RINE; (2) CoDE and SuSy are less sensitive to these manipulations;

(3) NPR shows improved performance under most kinds of manipulation. The inconsistent behavior of the detectors under simple image manipulations indicates the difficulty in developing a universal attack against different AIGI detectors.

From another perspective, these intervention experiments on the low-level image characteristics suggest that they do not have a direct and uniform causal effect on the accuracy of different detectors, despite the correlations witnessed in Table 1a. Future works could further investigate the underlying reasons for the correlations between these image characteristics and the detector accuracy.

Table 5: Detector accuracy (%) under different image manipulations. (The red/green numbers indicate decreased/increased accuracy compared to the baseline; underlined numbers indicate significant changes in accuracy of over 10%.)

| Manipulation | Factor | DRCT (ConvB) | DRCT (CLIP) | RINE | CoDE | SuSy | NPR |
|---|---|---|---|---|---|---|---|
| None | N/A | 65.6 | 83.1 | 31.8 | 66.7 | 83.8 | 66.8 |
| Lightness | 0.5 | 51.8 | 55.8 | 20.9 | 68.0 | 84.9 | 89.2 |
| | 1.5 | 71.7 | 93.6 | 77.8 | 69.7 | 79.4 | 82.0 |
| Contrast | 0.5 | 44.4 | 76.6 | 15.7 | 72.9 | 82.6 | 89.1 |
| | 1.5 | 76.3 | 93.3 | 84.1 | 69.5 | 85.5 | 87.3 |
| Saturation | 0.5 | 63.2 | 84.3 | 35.8 | 64.9 | 84.7 | 64.2 |
| | 1.5 | 67.6 | 90.9 | 56.7 | 65.1 | 82.7 | 68.5 |
| Sharpness | 0.5 | 76.6 | 85.3 | 34.0 | 71.2 | 84.9 | 89.5 |
| | 1.5 | 55.0 | 81.7 | 52.2 | 62.5 | 80.6 | 73.4 |

### B.5. The Relation Between Accuracy and Quality for Real Images

While this paper aims to investigate whether high-quality AI-generated images are more difficult to detect and understand the correlation between detector accuracy and AIGI quality, it is natural to ask whether similar conclusions hold for real images. To this end, we transfer the investigations to our collected real images. The results presented in Figure 13 do not show a similar consistent correlation between high detector accuracy and high quality scores across different preference models. It requires further studies to explore which kind of real images are easier or harder to be identified for AIGI detectors.

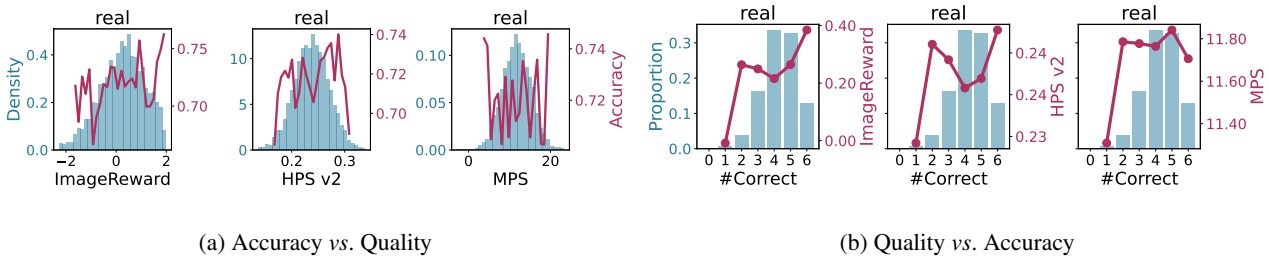

(a) Accuracy *vs*. Quality           (b) Quality *vs*. Accuracy

Figure 13: Investigating the relation between average detector accuracy and quality scores for **real** images. In contrast with the observations on fake images (Figures 6 and 7), the correlation between detector accuracy and image quality is less significant and relatively inconsistent across different human preference models.

## C. Qualitative Examples

We present random examples of our data corresponding to the short, medium-length, and long prompts in Figures 14 to 16. For each generated image, we also provide the average accuracy of detectors (*i.e.*, the proportion of detectors with correct predictions) and the quality scores in these figures.

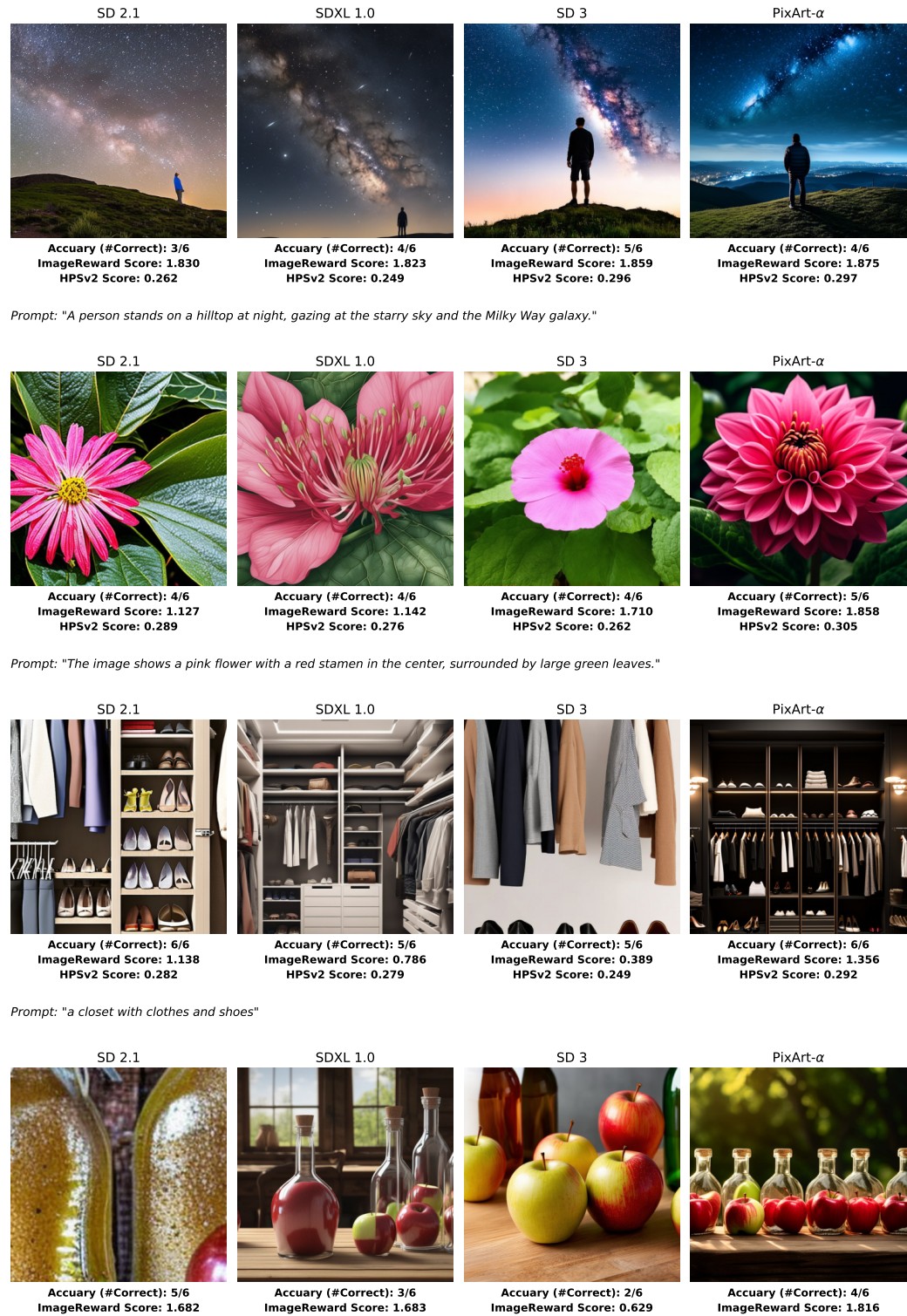

Figure 14: Random examples from our dataset corresponding to the **short** prompts (1-20 words).

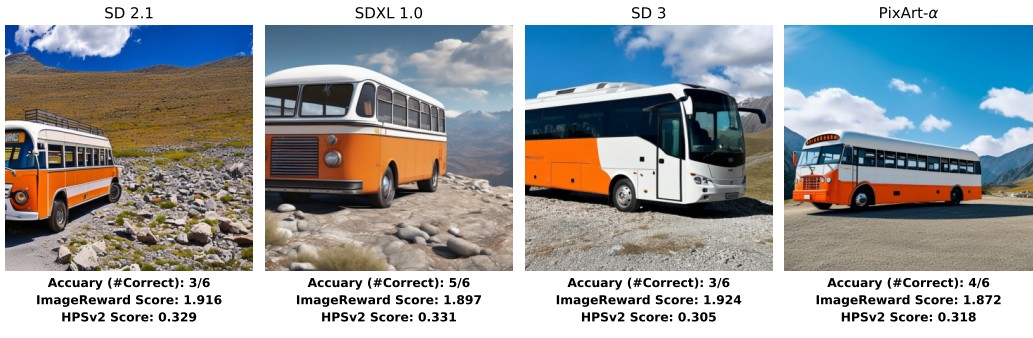

SD 2.1      SDXL 1.0      SD 3      PixArt-α

| Accuary (#Correct): 3/6 | Accuary (#Correct): 5/6 | Accuary (#Correct): 3/6 | Accuary (#Correct): 4/6 |
| ImageReward Score: 1.916 | ImageReward Score: 1.897 | ImageReward Score: 1.924 | ImageReward Score: 1.872 |
| HPSv2 Score: 0.329 | HPSv2 Score: 0.331 | HPSv2 Score: 0.305 | HPSv2 Score: 0.318 |

*Prompt: "A white and orange bus is parked on a rocky, mountainous terrain with a clear blue sky and scattered clouds in the background."*

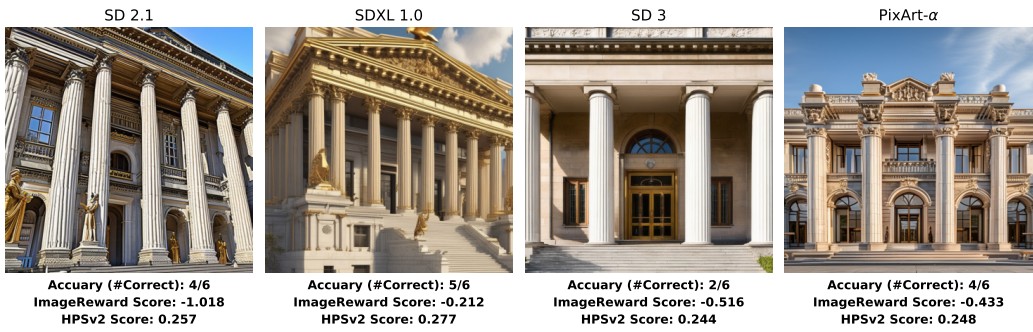

SD 2.1      SDXL 1.0      SD 3      PixArt-α

| Accuary (#Correct): 4/6 | Accuary (#Correct): 5/6 | Accuary (#Correct): 2/6 | Accuary (#Correct): 4/6 |
| ImageReward Score: -1.018 | ImageReward Score: -0.212 | ImageReward Score: -0.516 | ImageReward Score: -0.433 |
| HPSv2 Score: 0.257 | HPSv2 Score: 0.277 | HPSv2 Score: 0.244 | HPSv2 Score: 0.248 |

*Prompt: "The image showcases a grand, classical building with a prominent facade featuring tall columns, intricate stonework, and a decorative pediment. The building's roof is adorned with ornate details and a golden statue, while the entrance is flanked by large, detailed columns."*

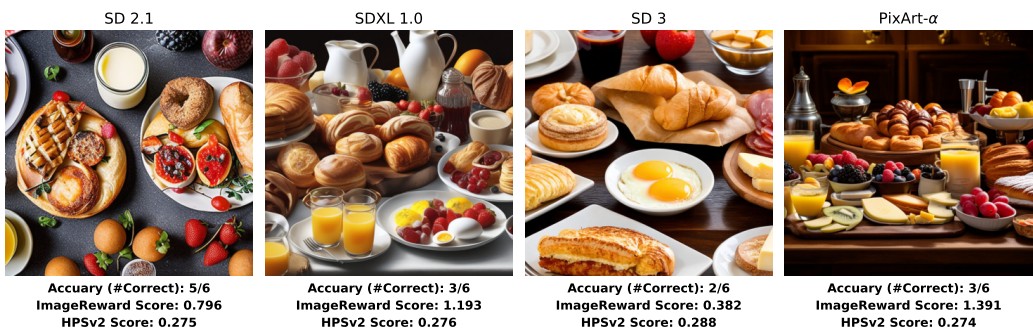

SD 2.1      SDXL 1.0      SD 3      PixArt-α

| Accuary (#Correct): 5/6 | Accuary (#Correct): 3/6 | Accuary (#Correct): 2/6 | Accuary (#Correct): 3/6 |
| ImageReward Score: 0.796 | ImageReward Score: 1.193 | ImageReward Score: 0.382 | ImageReward Score: 1.391 |
| HPSv2 Score: 0.275 | HPSv2 Score: 0.276 | HPSv2 Score: 0.288 | HPSv2 Score: 0.274 |

*Prompt: "The image depicts a lavish breakfast spread with an assortment of pastries, fruits, juices, eggs, meats, cheeses, and condiments, arranged on a table."*

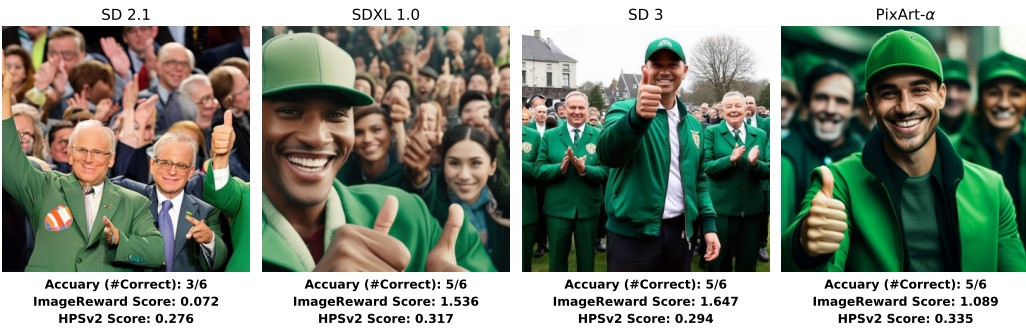

SD 2.1      SDXL 1.0      SD 3      PixArt-α

| Accuary (#Correct): 3/6 | Accuary (#Correct): 5/6 | Accuary (#Correct): 5/6 | Accuary (#Correct): 5/6 |
| ImageReward Score: 0.072 | ImageReward Score: 1.536 | ImageReward Score: 1.647 | ImageReward Score: 1.089 |
| HPSv2 Score: 0.276 | HPSv2 Score: 0.317 | HPSv2 Score: 0.294 | HPSv2 Score: 0.335 |

*Prompt: "The image depicts a man in a green jacket and cap, raising his right hand in a thumbs-up gesture, with a group of people in similar green attire clapping and smiling in the background."*

Figure 15: Random examples from our dataset corresponding to the **medium-length** prompts (21-40 words).

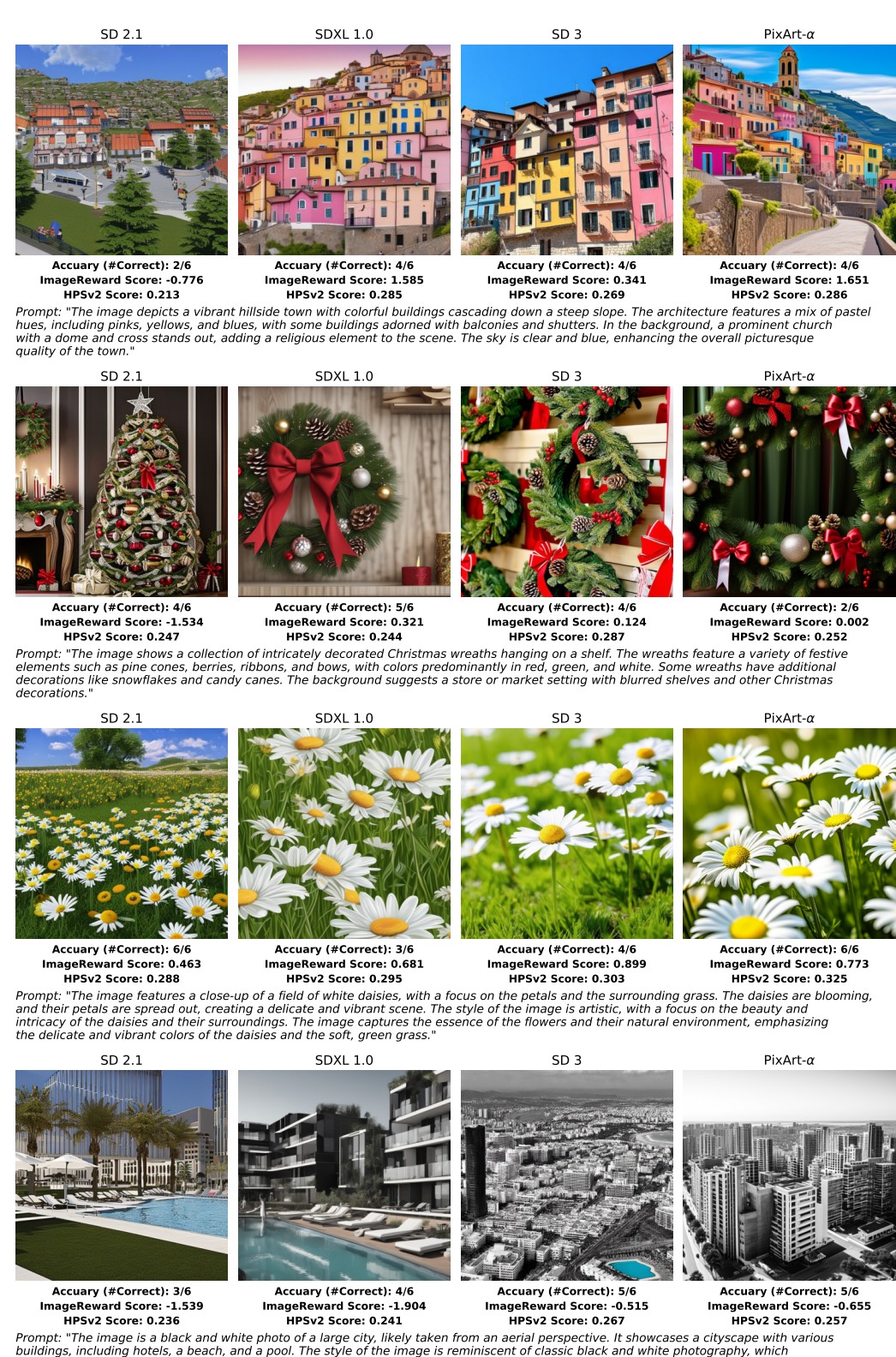

Figure 16: Random examples from our dataset corresponding to the **long** prompts (more than 40 words).

