# OpenReview forum: "Are High-Quality AI-Generated Images More Difficult for Models to Detect?"
_ICML.cc/2025/Conference — ICML 2025 poster_

### Official Review · Reviewer_auXT · 2025-03-10

**Overall Recommendation:** 3

**Summary:**

This paper investigates whether high-quality AI-generated images (AIGIs), as preferred by human perception models, are more difficult for detection models to distinguish from real images. Contrary to intuition, the authors find that images with higher human preference scores tend to be easier to detect by existing AIGI detectors.

**Claims And Evidence:**

Yes.

**Essential References Not Discussed:**

No.

**Experimental Designs Or Analyses:**

The experiments are well-designed and include appropriate baselines. Key strengths include:

1) Diverse dataset: Incorporates a variety of prompts, generators, and preference models;

2) Multiple detectors: Ensures robustness of findings across different architectures.

However, there are some potential concerns:

1) More details on dataset filtering would help: While the paper describes dataset construction, additional insights into how negative prompts and modifiers influence quality could strengthen the argument.

2) Further generalization studies: The study could benefit from testing whether similar findings hold for closed-source models like MidJourney and DALL·E 3.

**Methods And Evaluation Criteria:**

Yes.

**Other Comments Or Suggestions:**

1) Clarify how "negative prompts" impact image quality in dataset creation.

 The dataset includes negative prompts during generation to improve output quality, but it is unclear how these prompts influence image characteristics and whether they introduce biases in the evaluation.

 Do negative prompts primarily reduce artifacts, improve semantic fidelity, or affect low-level features like texture richness and contrast?

Could certain types of negative prompts inadvertently make images more detectable by standardizing certain visual properties (e.g., smoothing textures, removing unnatural edges)?

2) Consider testing adversarial modifications to verify detector robustness.

The study demonstrates that certain image characteristics (e.g., contrast, saturation, texture richness) correlate with detectability. However, adversarial image manipulations (e.g., adjusting contrast, adding noise, altering texture patterns) could be used to challenge these conclusions.

Testing adversarial perturbations (e.g., reducing texture richness in high-quality images) could reveal whether detectors are relying on spurious correlations or genuinely robust visual cues.

 Are there specific image transformations that can fool existing AIGI detectors while maintaining a high human preference score?

**Other Strengths And Weaknesses:**

1) Limited discussion on failure cases (e.g., when high-quality images are misclassified).

While the paper convincingly demonstrates that high-quality images tend to be more detectable, it does not provide an in-depth analysis of failure cases, i.e., scenarios where high-quality AI-generated images are misclassified as real or where lower-quality images fool the detectors.

Are there specific types of high-quality images (e.g., those with certain color distributions, structural complexity, or semantic attributes) that remain difficult to detect despite scoring highly on human preference models?

2) Evaluation on additional closed-source generators could enhance impact.

The study primarily focuses on open-source text-to-image models (e.g., Stable Diffusion variants, PixArt-α). However, many of the most widely used real-world AI-generated images come from closed-source models such as MidJourney, DALL·E 3, and Runway Gen-2.

**Questions For Authors:**

1) How do adversarial perturbations (e.g., reducing texture richness) affect detection accuracy?

2) Would a multi-task learning approach (joint training on quality scoring & detection) further improve detector performance?

3) Have you tested whether closed-source generators (e.g., MidJourney, DALL·E 3) exhibit the same quality-detectability trends?

**Relation To Broader Scientific Literature:**

Overall, the work fills a potential research gap by linking human preference-based quality scoring with AIGI detectability.

**Theoretical Claims:**

The paper presents no theoretical claims.

---

> ### Author Rebuttal · Authors · 2025-04-01
>
> **Q1: How do negative prompts and modifiers influence image characteristics and quality?**
>
> Firstly, we provide an ablation study, comparing the average quality scores of SDXL images with and without negative prompts and positive modifiers. The table below suggests that **prompt engineering improves the average quality of generated images**.
>
> |Negative prompts & positive modifiers|ImageReward|HPSv2|MPS|
> |---|---|---|---|
> |Without|0.0092|0.2452|12.14|
> |With|0.1172|0.2602|12.54|
>
> Since the modifiers are randomly sampled for each image, we compute the average accuracy and quality scores with and without each modifier for SD 3 images. The table below shows that none of these modifiers alone has significant and consistent effects on image quality or accuracy. We argue that the specific effect of modifiers may depend on the prompt and the synergy between modifiers.
>
> |Metric|Modifier|HDR|best quality|dynamic lighting|hyper detailed|photorealistic|professional lighting|ultra highres|ultra realistic|
> |---|---|---|---|---|---|---|---|---|---|
> |Accuracy (%)|Without|56.5|57.2|56.7|56.6|56.7|56.7|57.3|56.5|
> | |With|56.8|56.1|56.5|56.6|56.5|56.5|56.0|56.7|
> |ImageReward|Without|0.6123|0.6265|0.6515|0.6268|0.6076|0.6395|0.6245|0.6339|
> | |With|0.6429|0.6285|0.6035|0.6282|0.6476|0.615|0.6304|0.6212|
> |HPSv2|Without|0.2655|0.2662|0.2661|0.2648|0.2640|0.2657|0.2652|0.2649|
> | |With|0.2660|0.2652|0.2654|0.2666|0.2675|0.2658|0.2662|0.2665|
> |MPS|Without|12.93|13.00|13.00|12.99|12.95|13.12|12.93|13.04|
> | |With|13.13|13.06|13.06|13.07|13.11|12.94|13.13|13.02|
>
> **Q2: Test on closed-source models like MidJourney and DALL·E 3.**
>
> Please refer to Q4 of Reviewer rmyb.
>
> **Q3: Discussion on failure cases (e.g., when high-quality images are misclassified).**
>
> We take the high-quality images with top 30% ImageReward and HPS v2 scores, and regard the samples with at most 2 correct predictions for the 6 detectors (#Correct<=2) as failure cases, comparing them with the cases of #Correct>=5. We find that the failure cases consistently have lower average contrast and saturation, which is consistent with our conclusions in Sec. 3.
>
> |Metric|#Correct|SD 2.1|SD 3|SDXL 1.0|PA-alpha|FLUX.1 [dev]|Infinity|Average|
> |---|---|---|---|---|---|---|---|---|
> |contrast|>=5|66.19|67.46|53.72|63.96|60.44|58.86|61.77|
> | |<=2|51.11|61.58|49.60|45.72|52.67|51.62|52.05|
> |saturation|>=5|0.4343|0.4604|0.4551|0.5400|0.4817|0.4714|0.4738|
> | |<=2|0.2478|0.3553|0.4134|0.4043|0.4079|0.2982|0.3545|
>
>
> **Q4: Consider testing adversarial modifications to verify detector robustness.**
>
> Table 1(a) of the paper indicates certain adversarial directions for image modification, such as lowering the lightness/contrast/saturation, or increasing the sharpness. We implement these manipulations with different factors (0.5 means decreasing by 50%; 1.5 means increasing by 50%). The results below suggest that the adversarial modifications are effective for DRCT and RINE, while CoDE and SuSy are less sensitive to such modifications. Interestingly, NPR shows stronger performance on any manipulated data.
>
> |Manipulation|Factor|DRCT-ConvB|DRCT-CLIP|RINE|CoDE|SuSy|NPR|Avg|
> |---|---|---|---|---|---|---|---|---|
> |none|N/A|65.6|83.1|31.8|66.7|83.8|66.8|66.3|
> |lightness|0.5|51.8|55.8|20.9|68.0|84.9|89.2|61.8|
> | |1.5|71.7|93.6|77.8|69.7|79.4|82.0|79.0|
> |contrast|0.5|44.4|76.6|15.7|72.9|82.6|89.1|63.6|
> | |1.5|76.3|93.3|84.1|69.5|85.5|87.3|82.7|
> |saturation|0.5|63.2|84.3|35.8|64.9|84.7|64.2|66.2|
> | |1.5|67.6|90.9|56.7|65.1|82.7|68.5|71.9|
> |sharpness|0.5|76.6|85.3|34.0|71.2|84.9|89.5|73.6|
> | |1.5|55.0|81.7|52.2|62.5|80.6|73.4|67.6|
>
> **Q5: Are there specific image transformations that can fool existing AIGI detectors while maintaining a high human preference score?**
>
> To the best of our knowledge, there may not be simple transformations with these properties. However, one may consider replacing the perceptual constraint in the perceptual adversarial attack [a] with the preference score constraint and implement an effective attack for this purpose.
>
> [a] Perceptual Adversarial Robustness: Defense Against Unseen Threat Models. ICLR 2021.
>
> **Q6: Would a multi-task learning approach (joint training on quality scoring & detection) further improve detector performance?**
>
> We believe that this is possible as training on quality scoring could encourage the detector to learn on more quality-related features, especially those of higher levels, which could make the detector more robust to the distribution shift of low-level features in the cross-generator scenario.

---

### Official Review · Reviewer_ynZn · 2025-03-13

**Overall Recommendation:** 3

**Summary:**

This paper reveals an interesting yet counterintuitive phenomenon, where a higher-quality AI-generated image (AIGI) preferred by humans can be more easier for existing AIGI detectors to detect. The authors then investigate this effect and find that (1) images generated from short prompts and (2) certain image characteristics, such as texture richness, jointly influence both quality scores and detection accuracy. To address this, they propose a novel method to enhance the detection performance of existing patch-based detectors.

**Claims And Evidence:**

The authors have provided reasonable evidence to support their findings, and most claims proposed in the introduction section are well-supported by proper visualizations and references.

**Essential References Not Discussed:**

I recommend that the authors discuss similar research focusing on resolution, such as [1] and [2]. It would be valuable to provide a detailed comparison with these studies, highlighting the unique contributions of this paper to the field. This will help readers understand the distinct advancements and insights offered by this work.


[1] DF40: Toward Next-Generation Deepfake Detection, NeurIPS 2024; [2] Exploring Strengths and Weaknesses of Super-Resolution Attack in Deepfake Detection, ArXiv 2024; [3] A Quality-Centric Framework for Generic Deepfake Detection, ArXiv 2024.

**Experimental Designs Or Analyses:**

I have checked the soundness of the proposed experimental analyses, which all make sense to me.

**Methods And Evaluation Criteria:**

The benchmark datasets and evaluation criteria used are widely accepted in the field.

**Other Comments Or Suggestions:**

I don't have other comments in this section. Please see the question section and weakness section.

**Other Strengths And Weaknesses:**

Other Strengths:
- The originality of this paper is quite high and brings new insights to the field. I believe these findings actively and positively contribute to the entire field.
- The paper is well-written and very easy to follow. Most claims are well-supported by suitable evidence.

Other Weaknesses:
- While the findings have the potential to be applied to a broader range of fields, including face-swapping, face-reenactment, face-editing, and realistic image generation, the authors do not provide a comprehensive discussion on these potential applications.
- The paper is missing evaluations with more model architectures, which are necessary to verify the generality of the findings. Additional models such as CLIP, ViT (ImageNet), and CNNs trained from scratch should be included.
- The proposed method to address the resolution issue is somewhat limited, focusing primarily on patch-based detectors. There is potential for broader applicability beyond these specific detectors, which is a minor but notable weakness.
- The paper could benefit from using more recent and comprehensive datasets to make its evaluations more solid and robust.

**Questions For Authors:**

- Frequency Domain Analysis: Since resolution is strongly related to the frequency domain, could analyzing from the frequency domain provide some new insights?
- Detection of High-Quality Images: Would high-quality images, such as super-resolution images, be easier to detect? What are the reasons behind this? Could you provide further analysis? For instance, if face-swapping is performed and then followed by super-resolution, would this make the detection easier?
- Traces in High-Quality and Low-Quality Images: Theoretically, both high-quality and low-quality images contain traces of DNN generation models. Why are high-quality images easier to detect?
- Mitigating Overfitting on High-Quality Images: How can we alleviate the overfitting of detection models to the forgery traces in high-quality images and reduce the performance gap between different quality levels?

**Relation To Broader Scientific Literature:**

I believe the key contributions of this paper should not be limited to the detection of entire-image synthesis alone.

- Broader Application in Deepfake Detection: Similar phenomena have been observed in the detection of face-swapping content [1], where it has been noted that a resolution gap can lead to model shortcuts and overfitting, thereby limiting generalization. The findings of this paper have the potential to be applied in other related fields, such as deepfake detection, not just the detection of text-to-image generation content.
- Enhancing the Generation Process: Beyond improving detection, this technique can also benefit the generation process. The detectors can serve as critic models or reward models, helping to generate more realistic and undetectable AI-generated images (AIGIs). This dual application of the paper's findings could have significant implications and connections to other scientific literature.

[1] DF40: Toward Next-Generation Deepfake Detection, NeurIPS 2024.

**Theoretical Claims:**

N/A.

---

> ### Author Rebuttal · Authors · 2025-04-01
>
> **Q1: It is recommended to discuss similar research focusing on resolution, such as [1,2].**
>
> Thank you for your valuable suggestion. While high resolution is usually an important aspect of "high-quality" images in a broad sense, this paper considers a narrower sense of image quality, which is evaluated by human preference models. The generated images studied in this paper generally have the same resolution for each generator, while [1,2] emphasize the discrepancy between low-resolution and high-resolution deepfake images, as well as the influence of super-resolution. The paper [3], as you mentioned, focuses on "forgery quality" (i.e., *"whether the deepfake image is realistic or not"*), instead of the blurriness or resolution of images, which is closer to our definition of quality. We will include more discussions on these related works and clarify our focus.
>
> **Q2: This paper does not provide a comprehensive discussion on potential broader applications.**
>
> As you kindly suggested, the methodology of this research could be extended to related tasks such as image manipulation detection, and the results in this paper may benefit the research on improving the generative models or building more robust detectors. We will add these discussions to our revision.
>
>
> **Q3: The paper is missing evaluations with more model architectures, such as CLIP, ViT (ImageNet), and CNNs trained from scratch.**
>
> We agree that evaluation with diverse model architectures is important for reaching a reliable conclusion. The experiments in Sec. 5 already included detectors with different architectures, such as CLIP (RINE, DRCT), ImageNet-pretrained ViT (CoDE), and ResNet trained from scratch (NPR). Besides, we add more evaluations of the quality-accuracy correlation on generators of different architectures (please refer to Q4 of Reviewer rmyb).
>
> **Q4: The proposed method to address the resolution issue is somewhat limited, focusing primarily on patch-based detectors.**
>
> We acknowledge that the proposed patch selection strategies are restricted to patch-based detectors. However, other kinds of detection methods could also benefit from the results of this paper. For example, our regression models obtained in Sec. 4.3 could be applied to identify the hard samples for detection, and emphasizing these samples in model training may enhance the generalization.
>
>
> **Q5: The paper could benefit from using more recent and comprehensive datasets.**
>
> Thank you for your advice. In our response to Q4 of Reviewer rmyb, we validate our main conclusions on recent datasets of real-world generated images from commercial models. As for the experiments in Sec. 5, DRCT-2M is the latest published comprehensive benchmark for AIGI detection to the best of our knowledge.
>
>
> **Q6: Since resolution is strongly related to the frequency domain, could analyzing from the frequency domain provide some new insights?**
>
> Certain fingerprints of fake images such as the up-sampling traces utilized by NPR (Tan et al., 2024) may be witnessed in the frequency domain. Nonetheless, our preliminary experiments suggest that frequency domain analysis may not bring meaningful insights into the image quality we focus on.
>
> **Q7: Would high-quality images, such as super-resolution images, be easier to detect?**
>
> Thank you for raising this valuable question. Since generative super-resolution with diffusion models is popular in real-world applications, we collect 1000 fake images generated by an SDXL variant without or with generative super-resolution ($1024\to1536$). We test the detectors on these images, and the results below suggest that whether the super-resolution images are easier or harder to detect depends on the detector. We believe that this question is worthy of further study.
>
> |Generative super-resolution|CoDE|DRCT-ConvB|DRCT-CLIP|NPR|RINE|SuSy|avg|
> |---|---|---|---|---|---|---|---|
> |Without|76.0|74.6|51.4|100.0|59.7|78.8|73.4|
> |With|57.3|88.3|65.2|99.9|27.5|75.4|68.9|
>
> **Q8: Why are high-quality images easier to detect, given that both high-quality and low-quality images contain traces of DNN generation models?**
>
> According to our analyses in Sec. 4, existing detectors are sensitive to certain features that correlate with high quality scores, such as high contrast and high texture richness, which means the traces of generation may be more prominent in such images for the detectors.
>
> **Q9: How can we alleviate the overfitting of detection models to the forgery traces in high-quality images and reduce the performance gap between different quality levels?**
>
> We may collect more diverse and balanced data in terms of image quality. Moreover, designing models that are more capable of detecting higher-level artifacts like the distortion of object structures could reduce the performance gap, as such artifacts are common in low-quality images.

---

> > ### Comment · Reviewer_ynZn · 2025-04-04
> >
> > Thanks for the rebuttal. I think the Q7, Q8, and Q9 are very important problems for future research in AIGI detection. We hope the author could provide an in-depth and further discussion here. Overall, the authors have addressed most of my initial concerns so I maintain my initial rating.

---

> > > ### Author Response · Authors · 2025-04-09
> > >
> > > We sincerely thank you for your support and highlighting the importance of Q7, Q8, and Q9. We acknowledge that these raise significant and valuable directions for future research in AIGI detection. However, considering the limited time of the rebuttal and the scope of the paper, a more in-depth discussion of these complex issues is not feasible at this time. We recognize the importance of these questions and plan to address them in our future work.

---

### Official Review · Reviewer_581z · 2025-03-17

**Overall Recommendation:** 3

**Summary:**

This work considers the relationship between AI-generated images and real images, noting a counterintuitive phenomenon: generated images with higher quality scores, as assessed by human preference models, tend to be more easily detected by existing AIGI detectors. Additionally, it is observed that images generated from short prompts tend to achieve higher preference scores while being easier to detect.

**Claims And Evidence:**

The authors illustrate this phenomenon using a distribution plot (e.g., Fig1), but the relationship between the Accuracy curve and the distribution is not clearly explained. As a result, readers may find it challenging to quickly identify and interpret the correspondence between the accuracy and the underlying distributions of image quality scores.

**Essential References Not Discussed:**

No.

**Experimental Designs Or Analyses:**

I believe that while the authors successfully summarize phenomena through histogram distributions, the underlying reasons are not well explained. Additionally, the descriptions accompanying the histograms lack clarity, potentially leading to confusion among readers.

**Methods And Evaluation Criteria:**

The authors specifically collect an AIGI dataset to evaluate image quality and the difficulty of detection, utilizing a human preference model for assessment. Specifically, the authors utilize two pre-trained human preference models and six existing open-source AIGI detectors to support their experimental findings.

**Other Comments Or Suggestions:**

Please refer to the weakness.

**Other Strengths And Weaknesses:**

Strengths:

1.The paper is well-written and clearly structured.

2.The authors focus on a meaningful research direction, especially the  detection of AI-generated images.

3.The experimental validation is comprehensive.

Weaknesses:

1.The data histograms provided by the authors are not sufficiently intuitive. Specifically, the relationship between the distribution and the scores lacks clarity.

2.The experiments primarily discuss observations and phenomena but lack a deeper theoretical analysis or exploration of underlying causes.

**Questions For Authors:**

Do the authors provide comprehensive statistics on the dataset construction, such as the quantity and distribution of each category? Additionally, is there a clear indication of whether the dataset will be publicly available in the near future?

**Relation To Broader Scientific Literature:**

This paper reveals that high-quality AI-generated images (AIGIs), as preferred by humans, tend to be easier for existing AIGI detectors to identify. To further investigate this phenomenon, this paper analyzes how text conditions and image features influence the correlation between the average accuracy of detectors and the quality scores predicted by human preference models.

**Theoretical Claims:**

The authors primarily rely on extensive experiments to support their findings, but the study lacks sufficient theoretical analysis.

---

> ### Author Rebuttal · Authors · 2025-04-01
>
> **Q1: The relationship between the curve and the histogram in the plots is not clearly explained.**
>
> Thank you for pointing out the potential difficulty for readers to understand the figures. In Figure 1-3, the red curve illustrates how variable $y$ (e.g., the accuracy in Fig. 1) changes with respect to $x$ (e.g., the quality score in Fig. 1), and the blue histogram depicts the distribution of $x$. The curve and the histogram share the same $x$-axis.  We will revise the captions and related descriptions of Figure 1-3 to improve the clarity.
>
> **Q2: The underlying reasons for the phenomena illustrated by the figures are not well explained.**
>
> The main observation illustrated by Fig. 1 (i.e., generated images of higher quality tend to be easier to detect) **can be explained by our analyses in Sec. 4**. To clarify, we do not suppose that the relation between the detector accuracy and the quality scores is causal; instead, we aim to explore the **potential confounders underlying the counterintuitive positive correlation** between the two variables. Our results in Sec. 4.3 suggests that certain low-level image characteristics and high-level features may be the confounders for this correlation, and our experiments in Sec. 5 further validate that these low-level image characteristics can be utilized to predict the detectability of image patches on a broader range of data. We will make these points clearer in the revision.
>
> **Q3: This study lacks a deeper theoretical analysis or exploration of underlying causes.**
>
> We understand that theoretical analysis can provide deeper insights into the relationship between the detector accuracy and the image quality, although this paper is supposed to be an empirical study. We will provide a discussion from the causal perspective (as described in Q2) with a theoretical causal graph that explicitly depicts the relations among the variables we studied in this paper.
>
> **Q4: Provide comprehensive statistics on the dataset construction, such as the quantity and distribution of each category.**
>
> Thank you for your kind suggestion. The distributions of the image quality scores and the prompt lengths are depicted by the histograms in Fig. 1 and Fig. 3, respectively. However, different from some previous datasets for AIGI detection that are based on certain object categories (e.g., ForenSynths (Wang et al., 2020) and GenImage (Zhu et al., 2023)), the images in our dataset contain diverse content including complex scenes with various objects. Therefore, it is difficult to categorize the images and provide reliable statistics on the image content.
>
>
> **Q5: Will the dataset be publicly available in the near future?**
>
> Yes. As stated in the footnote on Page 3 of the paper, the dataset will be publicly available upon acceptance. More specifically, we will release the generated images, the corresponding prompts, and other metadata such as the denoising steps for diffusion models and the JPEG compression quality.

---

> > ### Comment · Reviewer_581z · 2025-04-04
> >
> > Thanks for response. I will maintain my rating.

---

### Official Review · Reviewer_rmyb · 2025-03-17

**Overall Recommendation:** 3

**Summary:**

The paper investigates the detectability of AI-generated images (AIGIs), revealing a counterintuitive finding: higher-quality AI-generated images, preferred by humans, are actually easier for detection models to identify. It shows that prompt complexity (with shorter prompts producing higher quality, more detectable images) and specific image characteristics (like high saturation, contrast, and rich textures) significantly influence both image quality scores and detector accuracy. Finally, the paper demonstrates how these insights can be practically leveraged to enhance the performance of existing detectors by optimizing input patch selection based on predicted detectability.

**Claims And Evidence:**

Even though the paper is very interesting, several of its claims lack clear substantiation:

1. On lines `045-052`, the paper claims that existing datasets are randomly generated without ranking, leading to discrepancies between training data and real-world applications. However, existing literature [1][2][3] shows that some detectors perform well even with out-of-distribution data, which appears to conflict with the paper's claim. The authors should address this discrepancy explicitly.

2. The paper argues (lines `110-111`) that existing datasets lack diversity but doesn't provide evidence for that. Moreover, prior studies [1][2][3] already include diverse samples generated from both GAN-based and diffusion-based models, including Stable Diffusion used by the authors themselves.

3. The paper mentions the lack of advanced generators in existing datasets. However, current benchmarks already include both GAN and diffusion-based models, specifically Stable Diffusion, which is also utilized in this paper.

4. The study employs only **four** generators, three of which (Stable Diffusion variants) likely have similar architectures, training datasets, and methodologies. This limitation could bias the results. Including a wider variety of generators, particularly those with differing architectures (e.g. Dall E 3, Imagen 3, Flux -- both open and close source models), would strengthen the validity of the findings.

5. Importantly, the pre-trained detectors evaluated in this paper might be biased towards short prompts since they were likely trained on shorter prompt samples, given that the authors claim their dataset is the first to include high-quality images generated from longer prompts. This potential bias would significantly undermine the paper's conclusions. Therefore, providing quantitative evidence demonstrating that detector performance is not influenced by prompt length is imperative.

# Ref:
- [1] ArtiFact: A Large-Scale Dataset with Artificial and Factual Images for Generalizable and Robust Synthetic Image Detection
- [2] Forgery-aware Adaptive Transformer for Generalizable Synthetic Image Detection
- [3] Towards Universal Fake Image Detectors that Generalize Across Generative Models

**Essential References Not Discussed:**

[1] ArtiFact: A Large-Scale Dataset with Artificial and Factual Images for Generalizable and Robust Synthetic Image Detection
[2] Forgery-aware Adaptive Transformer for Generalizable Synthetic Image Detection
[3] Towards Universal Fake Image Detectors that Generalize Across Generative Models

**Experimental Designs Or Analyses:**

Yes

**Methods And Evaluation Criteria:**

Yes

**Other Comments Or Suggestions:**

The paper needs clarifications:

1. The paper claims that short prompts have higher quality and higher detectability. However, it also discusses the shortest prompt, which exhibits opposite characteristics. This contradiction is confusing for the authors. Intuitively, both short and long prompts should perform poorly, while a medium-length prompt, which hits the sweet spot, should perform well. Therefore, the authors are requested to resolve this inconsistency.

2. The authors consider prompts from datasets like COCO to be short, which leads to confusion regarding what is classified as short or long. The paper should provide qualitative examples of short, medium, and long prompts, along with their generated images and corresponding results.

3. The paper also needs to provide qualitative evidence (prompt with generated images) on how prompt length affects the generation results. Currently, only quantitative results are presented.

**Other Strengths And Weaknesses:**

No

**Questions For Authors:**

Check claim & evidence section

**Relation To Broader Scientific Literature:**

Yes

**Theoretical Claims:**

No theoretical claims

---

> ### Author Rebuttal · Authors · 2025-04-01
>
> **Q1: Discrepancy between existing datasets and real-world applications.**
>
> The discrepancy between existing datasets and real-world applications lies in many aspects, such as semantics, quality, and image compression. This paper focus on the quality (as explained in lines 110-117, left column), while some previous studies emphasize that the real-world performance of detectors can be affected by the image compression bias in existing datasets.
>
> We acknowledge that some detectors [1,2,3] are reported to perform well on some OOD data. However, we find that they still have poor performance on other datasets, as suggested by the results in: https://anonymous.4open.science/r/ICML2025-rebuttal-4584/table.pdf. We did not test the method proposed in [1] as it is not open-sourced.
>
> **Q2: Evidence for lacking diversity.**
>
> Thank you for pointing this out. To clarify, prior studies (such as [1,2,3] as you mentioned) collect images generated by diverse generators to study the cross-generator generalization of detectors. In this paper, we instead focus on the data diversity corresponding to the same generator, especially the diversity of (1) quality-related image features and (2) prompt complexity. Specifically:
>
> (1) The diversity of quality-related features is achieved by the independently sampled positive modifiers, which are not applied in most existing datasets.
> (2) In contrast to Synthbuster, which has the highest diversity in prompt complexity but 98.5% of its prompts are under 60 words, our dataset (Fig. 3) exhibits a significantly wider distribution of prompt lengths.
> We will revise the related descriptions in Sec. 2 and Sec. 3.1 to improve the clarification on dataset diversity.
>
> **Q3: The role of "advanced generators".**
>
> We agree that current benchmarks already contain diffusion-based models like SD (1/2/XL series). However, by "advanced generators", we refer to those more capable of producing high-quality images, especially DiTs. SD 3 and PixArt-α
>  are selected as representatives of DiTs, and the average quality of their generated images is higher than SD 2.1/XL with U-Net architecture, as indicated by the histogram of Fig. 1. We will improve the related descriptions and provide more statistical comparisons.
>
> **Q4: Results with more generators.**
>
> Thank you for your valuable suggestion. We supplemented our data with images generated by the commercial DiT model FLUX.1 [dev], and Infinity, an autoregressive model. We reproduce Fig. 1/2 on the extended data and the results are presented in Fig. I/II in [a]. In addition, we validate our findings on closed-sourced models Midjourney v6 and DALL·E 3 based on images sampled from existing datasets in Fig. V/VI in [a]. The phenomena in these figures are consistent with Fig. 1/2 of the paper.
>
> [a] https://anonymous.4open.science/r/ICML2025-rebuttal-4584/quality_and_accuracy.pdf
>
> **Q5: The biases towards short prompts.**
>
> We agree that these pre-trained detectors might be biased towards images generated from short prompts if their training data only comprises such images. Fig. 3 does suggest that these detectors tend to have higher performance if the prompt length is below 40.
>
> However, the effects of prompt length on the detector accuracy are indirect. Specifically, longer prompts may include more objects and depict a more complex scene, or they could be more likely to mention certain attributes of objects that affect their visual appearance. Therefore, the emphasis of our analyses (Sec. 4.2/4.3/5) is on the visual characteristics of generated images instead of the prompts, and our results in Sec. 5 suggest that our conclusions can be utilized to improve the performance of detectors on existing datasets based on short prompts. Please refer to Q2 of Reviewer 581z for further explanations.
>
> **Q6: Images generated from the shortest prompt exhibit opposite characteristics.**
>
> Thank you for underlining the importance of this intriguing phenomenon. As explained in lines 210-215 (right column) and Appendix B (lines 694-729), we notice that the a significantly lower ratio of the shortest prompts contain color-related descriptions, while such descriptions tend to increase the saturation of image. Hence, compared with medium-length prompts (21-40 words), the shortest prompts induce lower saturation, which indicates lower accuracy according to Sec. 4.3. Therefore, this phenomenon does not contradict with our further analyses concerning image characteristics.
>
> **Q7: Explain the classification and provide qualitative examples of short, medium, and long prompts, along with their generated images and corresponding results.**
>
> Thank you for your advice. We will revise the descriptions related to prompt lengths according to the following classification standard: "short" means 1-20 words; "medium" means 21-40 words; "long" means more than 40 words.
>
> We provide randomly sampled qualitative examples accordingly in https://anonymous.4open.science/r/ICML2025-rebuttal-4584/qualitative.pdf.

---

### Official Review · Reviewer_gsPf · 2025-03-19

**Overall Recommendation:** 3

**Summary:**

This paper studied the correlation between the quality score of AI-generated images and the detection accuracy of AI-generated images. They found that AI-generated images with higher quality scores are easier to be detected by models. Then, they analyzed the influence of the length of text prompts and image quality features to the correlation. They also conducted experiments to study how to select patches for AI-generated image detection.

**Claims And Evidence:**

This paper tried to study and explain the correlation between the quality of AI-generated images and the detection accuracy of AI-generated images. However, the quality of image is evaluated by some pre-trained preference models, not by human. Since the image quality assessment is a highly subjective and challenging task, it is questionable that the quality scores given by these models can reflect actual human preference/image quality. In other words, are the images with high scores really high quality?

**Essential References Not Discussed:**

No.

**Experimental Designs Or Analyses:**

See “Methods And Evaluation Criteria”

**Methods And Evaluation Criteria:**

When we detect AI-generated images, only the image itself can be accessed. Therefore, the quality of an image should also be decided only by the image itself, why the quality score is determined by both text prompt and image? The correlation between the quality and the detection accuracy of AI-generated images observed in this paper maybe mainly influenced by the text prompt not by the image true quality.

**Other Comments Or Suggestions:**

Xu et al., 2024 “Imagereward: Learning and evaluating human preferences for text-to-image generation”This paper was published in NeurIPS 2023, not 2024.

**Other Strengths And Weaknesses:**

--How about the correlation between the quality of real images and the detection accuracy?
--In Section 4.3, why perform regression analyses at the cluster level, but not image level?
--The main finding of this paper is that AI-generated images with high quality scores are easier to be detected. But, in Section 5, the study shows that selecting high quality patches is not always helpful for AI-generated image detection, which is conflict with the main finding.
Besides, the experimental results shows that carefully selecting different quality patches cannot bring consistent performance gain.

**Questions For Authors:**

See “Claims And Evidence”, “Experimental Designs Or Analyses” and “weakness”.

**Relation To Broader Scientific Literature:**

N/A

**Theoretical Claims:**

There is no theoretical claim.

---

> ### Author Rebuttal · Authors · 2025-04-01
>
> **Q1: It is questionable that the quality scores given by the pre-trained preference models can reflect actual human preference/image quality.**
>
> We agree that the preference models could not replace humans in assessing image quality, and it is difficult to predict the human preference on image quality due to its subjective nature. Therefore, we show the validity of our conclusions by the **consistent phenomenon among different generators and different preference models**.  We acknowledge that two preference models may be insufficient to support the consistency, hence, we complement the main results with an additional preference model, MPS [b], and other generators (please refer to Q4 of Reviewer rmyb). The consistent results in Figure I/II in [a] further validate our conclusions.
>
> [a] https://anonymous.4open.science/r/ICML2025-rebuttal-4584/quality_and_accuracy.pdf
> [b] Learning Multi-dimensional Human Preference for Text-to-Image Generation.
>
>
> **Q2: Why is the quality score determined by both text prompt and image?**
>
> Existing preference models commonly take the text prompt for generation as input because a high-quality image in practice should be not only visually appealing but also aligned with the text prompt (i.e., satisfying the intention of users). However, as you kindly suggest, this paper should focus on the visual quality of the generated image alone.
>
> To this end, we try to minimize the influence of the image-text alignment in the comparison of image quality by replacing the text input for the preference models: instead of the original prompt for generation, we use the **BLIP-2 caption of the generated image itself**, which is expected to be well-aligned with the image as evaluated by the preference models. The corresponding results are presented in Figure III/IV in [a].
>
> **Q3: The correlation between the quality of real images and the detection accuracy.**
>
> Thank you for your suggestion. We provide the results in Figure VII/VIII in [a], which suggests that there is no significant and consistent correlation between the quality and detection accuracy for real images, indicating that existing detectors may learn different features on real and fake images.
>
> **Q4: Why perform regression analyses at the cluster level but not image level in Sec. 4.3?**
>
> At the image level, the accuracy is discrete (0/6, 1/6, ..., 6/6) as only 6 detectors are evaluated, which could induce overly high variance of the data and hinder the linear regression analyses. Therefore, we use the cluster-level data, where each sample is representative of a group of images with similar characteristics, to ease the regression analyses.
>
> **Q5: Sec. 5 shows that selecting high-quality patches is not always helpful for AI-generated image detection, which is in conflict with the main finding.**
>
> We acknowledge that the proposed patch selection strategy may not always bring a performance gain for different data and different detectors. This is expected as our main observation (i.e., generated images with higher quality scores *tend to* be easier to detect) is **statistically** valid, and its application to the improvement of detectors may depend on the characteristics of the detector and certain implementations such as the image patchification strategy. We believe that the findings of this paper could motivate future studies to propose more effective detectors with improved algorithmic designs.
>
> **Q6: The ImageReward paper was published in NeurIPS 2023, not 2024.**
>
> We are grateful for your careful reading! We will correct this error and proofread the references in the paper.

---

> > ### Comment · Reviewer_gsPf · 2025-04-07
> >
> > Thanks for the rebuttal.  I have increased my rating.

---

### Decision · Program_Chairs · 2025-05-01

**Decision:**

Accept (poster)

**Comment:**

This paper received 5 weak accept ratings.

Before rebuttal,

Reviewers suggested the following strength of the paper: 1) paper is very interesting (reviewer rmyb); 2) paper is well-written and clearly structured (reviewer 581z, ynZn); 3) experimental validation is comprehensive (reviewer 581z, auXT); 4) The originality of this paper is quite high and brings new insights to the field (reviewer ynZn). The weakness of the paper are: 1) some conflicts in the paper (Reviewer gsPf, rmyb); 2) several of its claims lack clear substantiation (Reviewer rmyb);  3) lack a deeper theoretical analysis or exploration of underlying causes (Reviewer 581z); 4) The paper is missing evaluations with more model architectures (Reviewer ynZn); 5) The proposed method to address the resolution issue is somewhat limited, focusing primarily on patch-based detectors (Reviewer ynZn); 6) The paper could benefit from using more recent and comprehensive datasets to make its evaluations more solid and robust (Reviewer ynZn); 7) More details on dataset filtering would help (Reviewer auXT); 8) Further generalization studies (Reviewer auXT); 9) Limited discussion on failure cases (Reviewer auXT); 10) Evaluation on additional closed-source generators could enhance impact(Reviewer auXT).

After rebuttal, all reviewers confirmed that they have read the paper and would like to update the review if needed. Reviewer gsPf thought
authors addressed their concerns and increased the rating. Reviewer rmyb didn't provide additional comment and kept the rating. Reviewer 581z maintained their rating. Reviewer ynZn thought authors addressed most of their concerns and maintained the rating. Reviewer auXT also kept their rating.

Given these AC decided to give weak accept rating to this paper.